

# Photoemission "experiments" on holographic lattices

**Filip Herček[1,3], Vladan Gecin[2,3] and Mihailo Čubrović[3⋆]**

**1** Department of Physics, University of Novi Sad,
Trg Dositeja Obradovića 4, 21000 Novi Sad, Serbia
**2** Department of Physics, University of Belgrade,
Studentski Trg 12-16, 11000 Belgrade, Serbia
**3** Center for the Study of Complex Systems,
Institute of Physics Belgrade, University of Belgrade,
Pregrevica 118, 11080 Belgrade, Serbia

⋆ cubrovic@ipb.ac.rs

## Abstract

We construct a 2D holographic ionic lattice with hyperscaling-violating infrared geometry and study single-electron spectral functions ("ARPES photoemission curves") on this background. The spectra typically show a three-peak structure, where the central peak undergoes a crossover from a sharp but not Fermi-liquid-like quasiparticle to a wide incoherent maximum, and the broad side peaks resemble the Hubbard bands. These findings are partially explained by a perturbative near-horizon analysis of the bulk Dirac equation. Comparing the holographic Green functions in imaginary frequency with the Green functions of the Hubbard model obtained from quantum Monte Carlo, we find that the holographic model provides a very good fit to the Hubbard Green function. However, the information loss when transposing the holographic Green functions to imaginary frequencies implies that a deeper connection to Hubbard-like models remains questionable.

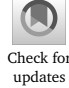

# 1  Introduction

Holography has established itself as a natural point of view in the subject of strongly correlated electron systems – the strong/weak duality translates the problem into tractable perturbative calculations in general relativity, and the relation of bulk physics to the renormalization group flow establishes a novel framework where many phenomena can be naturally expressed and understood [1–3]. Paradoxically, it is likewise indisputable that the field of AdS/cond-mat is still in its infancy in many respects. For example, controlled (top-down) constructions are few and quite complicated, thus we are mainly condemned to phenomenological, bottom-up models without explicit knowledge of the field theory Hamiltonian. Another issue is that the ultraviolet (UV) CFT puts rather strong constraints on physics at short distances, often quite unphysical in the context of condensed matter. Finally, the treatment of electrons on crystalline lattices – the bread and butter of condensed matter physics – is numerically demanding and has not yet become common among holographers. Here we aim to reduce this gap by constructing a holographic lattice with infrared (IR) geometry appropriate for a range of phases from normal to strange metals, and studying in detail the spectral functions (measured by angle-resolved photoemission spectroscopy (ARPES) in real life) of probe fermions on these backgrounds.

Spectral functions of probe bulk fermions on holographic lattices have been studied in some detail by now. The basic idea of true lattice models is to introduce a periodically modulated chemical potential (through the appropriate boundary condition for the bulk gauge

field), yielding a lattice of finite "hardness" in continuous space, akin to cold atoms in a periodic potential (or indeed the potential of a real-world crystal) and unlike the idealized lattice models like the Hubbard or $t-J$ model where the space itself is discrete. A perturbative analytical approach was pioneered in [4] and the full numerical solution (although for conductivity, not fermionic spectra) in [5, 6]. Afterwards, a wealth of research was performed. Limiting ourselves to the works on fermionic spectral functions, let us mention the exploration of the lattice Fermi surface in [7], the effects of striped phases on the fermionic spectra in [8–10], and the weakening and destruction of the Fermi surface due to anisotropy and nonlinear effects in [11, 12].

Some alternatives to the modulation of the chemical potential, known in the literature as the "ionic lattice", which requires solving numerically a system of nonlinear partial differential equations (PDE), are the Q-lattice which breaks translation invariance without actually introducing a lattice [13, 14] (see also [15]), the helical lattice in the Bianchi-VII background [16, 17] and the linear axion model [18]. We do not treat such approaches: the Q-lattice (despite the name!) and the linear axion are not really lattices and their Fermi surfaces do not live in Brillouin zones, whereas the Bianchi-VII geometry is a very special solution. We did however try out a number of approximations to the ionic lattice which go under the name of semiholography. The simplest, original idea [19] is to consider a free fermion on the lattice, coupled linearly to the probe bulk fermion whose propagator therefore acts as a self-energy in the propagator of the semiholographic theory. More elaborate setups are proposed in [20]. We have also solved the averaged equations of motion (essentially a multipole expansion of the lattice potential) as a convenient way to obtain a quick glance at the system or to perform large scans over the parameter space. It turns out that the spectra do not crucially depend on the approximation taken. Quite unexpectedly, even semiholography provides a decent quick and dirty way to compute the spectrum.

Running a bit forward, we will find that the lattice physics can be understood to a good extent by combining the near-horizon analysis of the spectral functions in homogeneous space [21–26] and the perturbative lattice calculations within the weak binding framework [4]. This is however only true for relatively weak lattices, where the amplitude of the chemical potential $\delta\mu$ is no larger than the mean $\mu_0$, i.e. $\delta\mu/\mu_0 \leq 1$. We are pretty much in the dark in the opposite, strong lattice regime. We realize the lattice in the hyperscaling-violating background of [27], a special case of the "effective holographic theories" or "holographic scaling atlas" framework of [28–31]. In homogeneous space, [27] shows that one can tune the parameters to have either a quasiparticle pole dressed in quantum-critical fluctuations, or just a quantum-critical continuum. On the lattice, the two regimes mix as different lattice momenta can belong to the first or the second regime. We will see that this provides an inhomogeneous and anisotorpic multi-peak structure of the spectral function (with quasiparticles, bands and wide bumps), which is not unlike the known phenomenology in real-world lattice models [32] and materials [33].

An important motivation and an additional challenge of this paper is the physics of the Hubbard model, the ubiquitous paradigm of non-Fermi liquids and strange metals, quite amenable to computational work at high temperatures and in certain approximations, yet still extremely rich, complex and unsolvable in the most general case (especially difficult at low temperatures), not the least because of the fermion sign problem [32, 34, 35]. No doubt the microscopy of this model has very little to do with the effective holographic theories – but the qualitative features of the spectrum are quite robust, and the general phenomenology as elucidated in [36–40] might be related to holographic physics.

Quantum Monte Carlo methods (specifically CTINT [41, 42]) provide us with high-quality Green functions of the Hubbard model at relatively high temperatures. We have directly

compared them to the holographic Green functions.[1] Therefore, a secondary goal of the paper is to find a holographic background (from the family of hyperscaling-violating geometries [27–29, 31]) which provides a good description of the correlation functions in the Hubbard model; one could then use the power of holography to study the low-temperature regime and other properties which are hard to access directly. It will turn out that we need one or two additional ingredients to model the Hubbard physics in AdS/CFT: regularization of the spectrum provided by a dynamical UV source, necessary to satisfy the ARPES sum rule [43], and perhaps the dipole coupling of the probe fermion (introduced first in [44, 45]) which makes the gaps in the spectrum more pronounced; we come close to the Hubbard model also without it but it does improve the fit. We emphasize however that the general phenomenology of holographic lattices and their spectra is important and interesting also in its own right: it yields a crossover from the quasiparticle regime to the strange metallic regime, and it will allow us to understand the effects of umklapp in the strong coupling regime.

The reader might be unhappy with the large number of figures and the emphasis on phenomenology as opposed to few equations and little analytical understanding. We certainly hope to gain a deeper view of our numerical findings, but in the present stage even a phenomenological exploration of the system is useful, hence the formulation "photoemission experiments" in the title, in conscious homage to [46].

The composition of the paper is the following. In section 2 we give the basic framework of our model: the Einstein-Maxwell-dilaton (EMD) system with hyperscaling violation and the ionic lattice; we describe the lattice solution in the bulk and list the relevant tunable parameters. Section 3 brings the Dirac equation, discussing the dipole term and the dynamical cutoff. Section 4 discusses in detail the properties of the spectral functions, the appearance of Hubbard-like bands and quasiparticles, and offers a tentative explanation for some phenomena through a perturbative analytic treatment. In Section 5, we explore how far the analogy to the Hubbard model can go by fitting our results (in imaginary frequency) to the quantum Monte Carlo results for the Hubbard Hamiltonian. In Section 6 we conclude the paper trying to find deeper implications of our results, in particular how fundamental the relation to the Hubbard model is and what to do next.

## 2 Holographic ionic lattice with hyperscaling violation

### 2.1 Action and geometry

The broad idea of the holographic setup is that a strange metal can be understood as a (not necessarily Fermi-liquid-like!) quasiparticle coupled to a bath of quantum critical excitations. This viewpoint (different in details and with a different choice of the holographic model) was pursued e.g. in [47] and formulated particularly clearly in the context of charge transport in [48]. To that end, we adopt from [27] a specific realization of the effective holographic theory of quantum critical phases of [29–31]. The holographic dual of this system is a theory with gravity, gauge and matter fields in $AdS_4$. Its action has the structure $S = S_{\text{bulk}} + S_{\text{bnd}}$, the sum of the bulk action and the boundary action. The boundary action contains the Gibbons-Hawking-York term and the regulator terms ($A_\mu \partial_\nu n^\nu A^\mu$ and $\Phi^2$) for the gauge and dilaton field respectively. We do not use $S_{\text{bnd}}$ explicitly (because we do not compute correlation functions of the EMD fields nor the thermodynamic quantities). The bulk action reads (in the $(-, +, +, +)$

---

[1] An important twist is that quantum Monte Carlo yields the Green function in imaginary (Matsubara) frequency, so we need to transform the holographic functions into imaginary frequencies too. Although this direction is easy – it is way harder to perform "analytical continuation" from imaginary to real frequencies – it will turn out nevertheless that comparing the Green functions in Matsubara frequencies destroys a lot of information and creates great difficulties.



convention that we use everywhere throughout the paper):

$$S = \int d^4x \sqrt{-g} \left[ R - (\nabla\Phi)^2 - Z(\Phi)F_{\mu\nu}F^{\mu\nu} + V(\Phi) \right].$$

(1)

Here, $F_{\mu\nu}$ is the electromagnetic field strength tensor and the potentials of the dilaton $Z$ and $V$ depend on some scaling exponents $\alpha$ and $\delta$:[2]

$$Z(\Phi) = \frac{1}{2}\left(1 + \cosh(2\alpha\Phi)\right), \quad V(\Phi) = 6 + V_0\left(1 - \cosh(2\delta\Phi)\right).$$

(2)

The important feature is that $Z$ and $V$ run as $\exp(2\alpha\Phi)$ and $\exp(2\delta\Phi)$ respectively in the IR and reduce to unity and 6 respectively in the UV of AdS$_4$ (we will show shortly that $\phi \to 0$ at the boundary). For all calculations in the paper we choose $V_0 = -0.5$. The charged black brane sources the gauge field of the form $A = A_t dt$. Ionic square lattice solutions are obtained by introducing a boundary source for the gauge field with the periodicity of a square lattice with period $1/2\pi Q$. For such solutions a numerically convenient choice of coordinates, suggested in [9, 10], is $(t, x, y, \tilde{z})$ where the AdS boundary is at $\tilde{z} = 1$ and the horizon is at $\tilde{z} = 0$. We denote by $r$ the usual radial coordinate which equals $\infty$ at the AdS boundary and some positive $r_h$ at the horizon, and its inverse is $z = 1/r$. The relation between the three radial coordinates is

$$\frac{r_h}{r} = \frac{z}{z_h} = 1 - \tilde{z}^2.$$

(3)

The $r$ coordinate will be convenient for analytical considerations near the horizon but the optimal coordinate choice for the numerical solution of EMD equations is $\tilde{z}$; the $z$ coordinate will be convenient when discussing the calculation of the spectral functions and their UV cutoff. The electrostatic potential at the AdS boundary for the square lattice with period $1/2\pi Q$ now reads

$$A_t(x, y, \tilde{z} = 1) = \mu(x, y) \equiv \mu_0 + \delta\mu\cos(2Q\pi x)\cos(2Q\pi y).$$

(4)

With this boundary condition the solution becomes inhomogeneous and the metric $g_{\mu\nu}$ as well as the gauge and matter fields $A_t$ and $\Phi$ depend on $x, y, \tilde{z}$ in the whole space. The most general form of the metric consistent with the symmetries of the square lattice is:

$$ds^2 = \frac{r_h^2}{(1-\tilde{z}^2)^2}\left[ -f(\tilde{z})q_{tt}(x,y,\tilde{z})dt^2 + q_{xx}(x,y,\tilde{z})\left(dx^2 + dy^2\right) + 2q_{xy}(x,y,\tilde{z})dxdy \right.$$
$$\left. + 2q_{x\tilde{z}}(x,y,\tilde{z})(dx+dy)d\tilde{z} + \frac{4\tilde{z}^2 q_{\tilde{z}\tilde{z}}(x,y,\tilde{z})}{r_h^2 f(\tilde{z})}d\tilde{z}^2 \right],$$

(5)

where $f(\tilde{z})$ is the redshift function (to be determined in the next subsection from the IR analysis) and $q_{tt}, q_{xx}, q_{xy}, q_{x\tilde{z}}, q_{\tilde{z}\tilde{z}}$ parametrize the inhomogeneous metric. Obviously $f$ could be absorbed into $q_{tt}$ and $q_{\tilde{z}\tilde{z}}$ but it is useful to keep it in order to understand the IR asymptotics. The gauge field is $A = A_t(x, y, \tilde{z})dt$ so the electric field strengths $E_x, E_y, E_z$ are all nonzero, and the dilaton is $\Phi = \Phi(x, y, \tilde{z})$.

### 2.1.1 IR asymptotics

In absence of lattice, this geometry was studied in detail in [27] and represents a case of hyperscaling-violating Lifshitz-scaling geometries [29–31] of the holographic "scaling atlas" [49]. With the lattice, it is extremely difficult to say anything about the IR geometry. It is

---

[2]These exponents determine the familiar Lifshitz exponent $\zeta$ and the hyperscaling-violation exponent $\theta$, discussed in many holographic publications, e.g. [29–31].

known that an "explicit", i.e. not spontaneously generated lattice is always irrelevant in far IR at sufficiently low temperatures; the study of holographic Mott insulators [8] has also found that only the spontaneously generated lattice survives at the horizon. However, for any specific bulk action, the question is how low the temperature should be, and how far from the boundary (i.e., how close to the extremal horizon) the lattice dies out. If the lattice dies out only slowly as we approach the horizon at relatively small (though nonzero) temperature, then we have no good idea what IR ansatz to take.[3] Assuming a "weak" lattice in the sense that the amplitude is no larger than the mean of the chemical potential:

$$\frac{\delta\mu}{\mu_0} \leq 1\,, \tag{6}$$

it makes sense to perform a perturbative expansion in $\delta\mu/\mu_0$ about a homogeneous scaling solution. The IR region is best described in the $r$ coordinate, hence we start from the IR ansatz

$$ds_{\text{IR}}^2 = -f(r)Q_{tt}(x,y,r)dt^2 + Q_{xx}(x,y,r)\left(dx^2 + dy^2\right) + 2Q_{xy}(x,y,r)dxdy$$
$$+ 2Q_{xr}(x,y,r)(dx+dy)dr + \frac{Q_{rr}(x,y,r)}{f(r)}dr^2\,. \tag{7}$$

The redshift function $f$ in (7) is the same as in (5), only in (5) we have expressed it in terms of $\tilde{z}$ rather than $r$. The other metric functions ($Q_{\mu\nu}$ vs. $q_{\mu\nu}$) differ in the two metrics and *a priori* have nothing to do with each other. The expansion around a scaling solution reads[4]

$$Q_{tt}(x,y,r\to 0) = r^{2\gamma}\mathcal{S}_{tt}\,, \quad Q_{xx}(x,y,r\to 0) = r^{2\beta}\mathcal{S}_{xx}\,, \quad Q_{rr}(x,y,r\to 0) = r^{-2\gamma}\mathcal{S}_{rr}\,,$$
$$Q_{xy}(x,y,r\to 0) = r^{2\nu}\mathcal{S}_{xy}\,, \quad Q_{xr}(x,y,r\to 0) = r^{2\lambda}\mathcal{S}_{xr}\,, \tag{8}$$
$$A_t(x,y,r\to 0) = r^{2\xi}f(r)\mathcal{S}_a\,, \quad \Phi(x,y,r\to 0) = 2\eta\log r\mathcal{S}_{\Phi}\,,$$

where the series $\mathcal{S}_{\text{field}}$ for any of the fields $Q_{tt}, Q_{xx}, Q_{xy}, Q_{xr}, Q_{rr}, A_t, \Phi$ has the form

$$\mathcal{S}_{\text{field}} = C_{\text{field}}^{(0)} + \sum_{n=1}^{\infty} r^n\Bigg[ C_{\text{field}}^{(n)} + D_{\text{field}}^{(n)}\cos(2Q\pi x)\cos(2Q\pi y) + E_{\text{field}}^{(n)}\cos(2Q\pi x)\sin(2Q\pi y)$$
$$+ F_{\text{field}}^{(n)}\sin(2Q\pi x)\cos(2Q\pi y) + G_{\text{field}}^{(n)}\sin(2Q\pi x)\sin(2Q\pi y) \Bigg]\,. \tag{9}$$

In other words, the leading-order scaling IR solution acquires both homogeneous and inhomogeneous corrections.

Among the zeroth-order terms $C_{\text{field}}^{(0)}$ in (9), we must have $C_{xy}^{(0)} = C_{xr}^{(0)} = 0$ as the off-diagonal metric terms are always zero (in the appropriate gauge) in the homogeneous limit. Picking the appropriate gauge we may always put $C_{xx}^{(0)} = 1$ and also $C_{tt}^{(0)} = 1/C_{rr}^{(0)}$. Also, $C_{\Phi}^{(0)} = 1$ because any non-unit value could be absorbed in $\eta$. Now equating the scaling exponents in the leading terms in each equation of the EMD system yields

$$\beta = \frac{(\alpha+\delta)^2}{4+(\alpha+\delta)^2}\,, \quad \gamma = 1 - \frac{2\delta(\alpha+\delta)}{4+(\alpha+\delta)^2}\,, \quad \eta = -\frac{2(\alpha+\delta)}{4+(\alpha+\delta)^2}\,,$$
$$\xi = \alpha\sqrt{\beta(1-\beta)} + \beta + \frac{5}{4}\,, \tag{10}$$

$$f(\tilde{z}) = 1 - \left(\frac{r_h}{r}\right)^{2\beta+2\gamma-1}\,, \quad T = \frac{C_{tt}^{(0)}(2\beta+2\gamma-1)}{4\pi}r_h^{2\gamma-1}\,. \tag{11}$$

---

[3]We thank Aristomenis Donos and Koenraad Schalm for their remarks on this issue.

[4]In principle we should be expanding starting from $r = r_h$ not $r = 0$ at any finite temperature, however we assume that the temperature is not very high and $r_h$ is reasonably small. Only in this case does it make sense to assume that an approximate scaling solution still exists, as argued also in [27].

This is just the homogeneous solution studied in [27] in slightly different notation. In order to have a smooth horizon the condition $2\gamma - 1 > 0$ has to be satisfied. We thus specialize to the domain $\delta > 0, \alpha < \alpha_0 < 0$ where $\alpha_0$ is a negative constant which can be expressed from $\alpha, \delta$ as in [27]. The coefficients of the first and higher order corrections can be obtained perturbatively in the usual way, equating with zero the prefactors of the terms in the small $r$ expansion of the equations of motion. We list the coefficients of a few leading terms in Appendix A. The important point is that all coefficients are zero at first order, and only at second order there is an inhomogeneous branch of the solution. Since the leading (and even the first subleading) term for all functions is homogeneous, we conclude that indeed the ionic lattice is irrelevant in IR. But this result is clearly perturbative as it hinges on expanding over the oscillatory part, hence it may be unjustified for strong lattices, with $\delta\mu/\mu_0$ large. Since we do not consider strong lattices in this paper, we adopt this solution as the IR boundary condition. The numerical solution is found to be convergent and consistent with these boundary conditions, further strengthening the result.

We are now ready to state the boundary conditions for the numerics. We expect no Maxwell hair at the horizon, so we impose Dirichlet boundary conditions for $A_t$, which should drop to zero, in accordance with (8). The behavior of the dilaton is also determined by (8): at the horizon it approaches some constant value which at leading (scaling) order equals $2\eta \log r_h$. This is not enough for stable numerics, so we have included also the subleading corrections as stated in (8) and Appendix A. The boundary condition is given by this IR expansion. In fact, in $\tilde{z}$ coordinate the outcome is very simple: the dilaton is smooth at the horizon and has zero derivative, i.e. in computational coordinates we have Neumann conditions.

Finally, the metric functions should all be smooth (the simple zero at the horizon is taken care of by the redshift function $f$), so we impose Neumann boundary conditions, requiring their derivatives to be zero, again with the falloff in accordance with the IR expansion.

### 2.1.2 UV asymptotics

The UV asymptotics are straightforward to obtain by expanding the equations of motion in the vicinity of $\tilde{z} = 1$:

$$q_{ii}(x, y, \tilde{z} \to 1) = 1 + \mathcal{C}_{ii}^{(2)}(1 - \tilde{z}^2)^2 + \dots,$$
$$q_{ij}(x, y, \tilde{z} \to 1) = \mathcal{C}_{ij}^{(2)}(1 - \tilde{z}^2)^2 + \dots,$$
$$A_t(x, y, \tilde{z} \to 1) = \mu(x, y) + a(x, y)(1 - \tilde{z}^2) + \dots,$$
$$\Phi(x, y, \tilde{z} \to 1) = \left(1 - \tilde{z}^2\right)^{\Delta_-}\left(1 + \mathcal{C}_{\Phi}^{(1)}(1 - \tilde{z}^2) + \mathcal{C}_{\Phi}^{(2)}(1 - \tilde{z}^2)^2 + \dots\right),$$
$$\Delta_- = \frac{3}{2} - \sqrt{\frac{9}{4} - 4\delta^2 V_0}, \tag{12}$$

where in the first equation the index $i$ can be $t$, $x$ or $\tilde{z}$ and in the second equation $ij$ is either $xy$ or $x\tilde{z}$, and the leading term $\mu$ in the expansion of the gauge field is the inhomogeneous chemical potential.[5] We do not list here the (cumbersome) explicit expressions for the series coefficients as finding them poses no principal problems and goes along the same lines as in [9–11]. The UV boundary conditions are the following: for the metric functions we need Dirichlet conditions to ensure the AdS asymptotics; for the scalar potential $A_t$ we likewise impose the Dirichlet condition with the boundary value (4). The boundary condition for the dilaton amounts to setting the leading term (proportional to the source) to unity, so that we do not have a spontaneous formation of the scalar condensate, therefore it is again of Dirichlet

---

[5]Note that the next term, $a(x, y)$, is not the charge density as the presence of the lattice makes the boundary integrals obtained in the on-shell action more complicated.

type; the effective mass squared of the dilaton field is $-4\delta^2 V_0$, from the form of the potential $V$ in (2).

The equations of motion are the standard EMD equations, altogether 7 in number – 5 independent components of Einstein equations, one Maxwell equation and one dilaton equation. This matches the 7 independent fields taking into account the $C_4$ symmetry of the square – $q_{tt}, q_{xx} = q_{yy}, q_{\tilde{z}\tilde{z}}, q_{xy}, q_{x\tilde{z}} = q_{y\tilde{z}}$, plus $A_t$ and $\Phi$. We do not give the equations in human-readable form as the expressions are huge (we keep them as Wolfram Mathematica files). In the next section we give a phenomenological overview of the numerics and the solutions obtained.

## 2.2 Square ionic lattice – the solution

We solve the equations of motion numerically, with the boundary conditions determined by the coefficients in the expansions (8-12). The resulting system of nonlinear PDEs is integrated by a collocation pseudospectral algorithm on Gauss-Lobatto grid along the radial direction $\tilde{z}$ and on Fourier grid along the transverse directions $x, y$. Our code is mainly based on [50, 51]. In Appendix B we give some more details on the integrator and some test examples. The production runs were performed on a lattice $2N \times 2N \times 2N$ with $N = 16$ but in the Appendix we have checked the convergence of the solution for varying lattice sizes from $N = 6$ to $N = 16$.

Typical solutions for the bulk are given in Figs. 1 and 2 for the matter and gauge fields and in Fig. 3 for components of the metric. The top panel of Fig. 1 shows that indeed the lattice is irrelevant: both fields modulate in UV but in IR the oscillations die out completely. The UV oscillations are of the same form $\cos(2Q\pi x)\cos(2Q\pi y)$ for both $A_t$ and $\Phi$, as we see in the bottom panel. Fig. 2 demonstrates that the scaling solution indeed survives in IR: both fields agree nicely with the analytical predictions for the scaling laws obtained in Eqs. (9-10). The figure is produced for a low temperature $T/Q = 0.1$ so the horizon radius is quite small; for higher temperatures the scaling is not as clear (expectedly, because the relevant scale is then $(r - r_h)/r_h$ rather than $r/r_h$). Finally, Fig. 3 confirms that the AdS asymptotics are preserved (no oscillations in UV) and that in deep IR the lattice also dies out.

All calculations in Section 4 are performed in this way, with the fully self-consistent PDE solver for the holographic ionic lattice. For mass runs needed for the comparison to the Hubbard model, we have also used approximate methods: averaged equations (multipole expansion) and semiholography (in the narrow sense, introduced in [19] and exploited e.g. in [52]) to gain a first guess at the solution; afterwards we have always confirmed the result through a full lattice calculation. The next two sections are devoted mainly to the full lattice calculation of the spectrum from the probe Dirac equation. In Appendix C we describe the other methods, which we all call semiholographic in the broad sense that one does not solve the full system of equations for the background.

# 3 Dirac equation for the probe fermion

## 3.1 The action and the equation of motion

We now come to the computation of the spectral functions. This is perhaps the "cleanest" way of doing bottom-up holography, as the outcome directly determines the density of states and the excitation spectrum of the electron in a given background. This calculation is well-established, both without [21, 22, 24, 25] and with a lattice [7–12]. The basic holographic dictionary entry stemming from the Gubser-Polyakov-Klebanov-Witten prescription [53, 54] relates the two-point propagator of the strongly coupled CFT electron to the probe Dirac fermion in AdS. Being a probe, it does not backreact on the other fields and propagates in the fixed

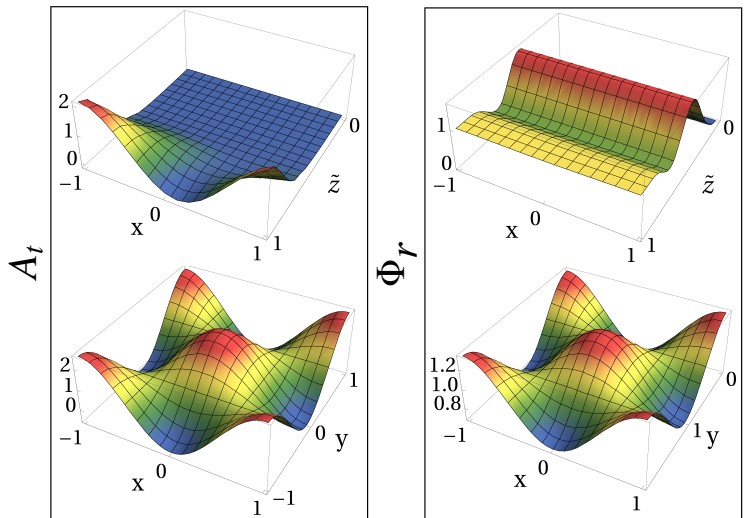

Figure 1: Numerically computed electrostatic field (left) and rescaled dilaton $\Phi_r \equiv \Phi/(1-\tilde{z}^2)$ (right), along an $x - \tilde{z}$ slice at $y = 0.5$ (top) and along an $x - y$ slice at the AdS boundary $\tilde{z} = 1$ (bottom). Both fields oscillate at the boundary but the oscillation amplitudes rapidly diminish toward the horizon because the ionic lattice is irrelevant in IR. Unlike the rescaled dilaton $\Phi_r$, the physical dilaton $\Phi$ does not oscillate at the boundary because the factor $1 - \tilde{z}^2$ kills the oscillations at $\tilde{z} = 1$. The scaling exponents are $(\alpha, \delta) = (-1.5, 1)$ at temperature $T = 0.1$, for $\mu_0 = 1$, $\delta\mu/\mu_0 = 1$ and $Q = 1/2$.

background found in Section 2 for given parameters $(\alpha, \delta)$. The probe fermion has charge $q$, mass $m$ and dipole momentum $\kappa$ (more on it later). The action again has the bulk and boundary term: $S_f = S_{f\,\text{bulk}} + S_{f\,\text{bnd}}$. The bulk action reads:

$$S_{f\,\text{bulk}} = i \int d^4 x \sqrt{-g}\, \bar{\Psi} \left[ \frac{1}{2} \left( \overleftarrow{\slashed{D}} + \overrightarrow{\slashed{D}} \right) - m - \frac{i}{2} \kappa \Gamma^{ab} e_a^\mu e_b^\nu F_{\mu\nu} \right] \Psi . \tag{13}$$

Here, $\Psi$ is the Dirac bispinor, $\slashed{D}$ is the covariant derivative determined by the spin connection $\omega_{ab\mu}$ and the gauge field $A_\mu$, and the last term is the coupling of the electron dipole momentum $\kappa$ with the field strength $F_{\mu\nu}$. Explicitly the covariant derivative reads

$$\slashed{D} = \Gamma^a e_a^\mu \left( \partial_\mu + \frac{1}{4} \omega_{bc\mu} \Gamma^{bc} - iq A_\mu \right) . \tag{14}$$

The indices $a, b, c \in \{t, x, y, z\}$ denote the local flat coordinates and $\Gamma^{ab} \equiv \left[ \Gamma^a, \Gamma^b \right]/2$. The expressions for the vielbein $e_a^\mu$ are quite cumbersome and we list them only as functions of a general metric $g_{\mu\nu}$ with nonzero components $g_{tt}, g_{xx} = g_{yy}, g_{rr}, g_{xy}, g_{xr}$. Inserting all the metric functions in their explicit form would result in expressions which are too cumbersome

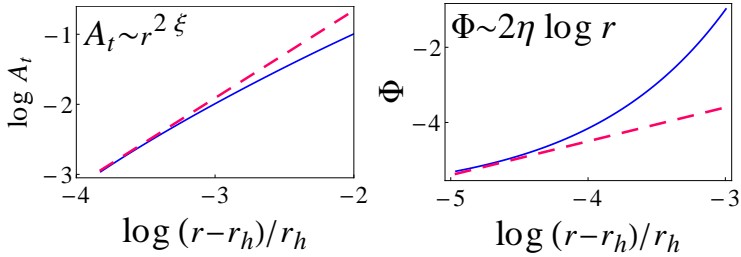

Figure 2: Far-IR scaling of the gauge field $A_t$ (left) and the dilaton $\Phi$ (right), as a function of the radial coordinate $r$ (related to $\tilde{z}$ as stated in Eq. (3)), for $(\alpha, \delta) = (-1.5, 1)$ at temperature $T = 0.1$, $\mu_0 = 1$, $\delta\mu/\mu_0 = 1$ and $Q = 1/2$. Full blue curves denote the numerical solution at $x = y = 0.5$ while the red dashed lines are the analytical scaling predictions from (9). Near the horizon the solutions are reasonably close to the scaling ansatz; further away they diverge. For higher temperatures the scaling is less and less prominent.

and not very illustrative:

$$e_t = \sqrt{-g_{tt}}\partial_t, \quad e_x = \sqrt{g_{xx} - g_{xy}}\partial_x + \sqrt{g_{xy}}\partial_r, \quad e_y = \sqrt{g_{xx} - g_{xy}}\partial_y + \sqrt{g_{xy}}\partial_r,$$

$$e_z = a_1\partial_x + a_2\partial_y + (a_3 + \sqrt{g_{rr}})\partial_r,$$

$$a_1 = g_{xr}\frac{\sqrt{g_{xx} - g_{xy}}}{g_{xx} - g_{xy}} + \frac{\sqrt{g_{xy}\left(-2g_{xr}^2 + g_{rr}\left(g_{xx} + g_{xy}\right)\right)}}{g_{xx} + g_{xy}},$$

$$a_2 = \sqrt{g_{xy}},$$

$$a_3 = \frac{2g_{xr}\sqrt{g_{xy}} - \sqrt{g_{rr}}\left(g_{xx} + g_{xy}\right) - \sqrt{g_{xx} + g_{xy}}\sqrt{-2g_{xr}^2 + g_{rr}\left(g_{xx} + g_{xy}\right)}}{g_{xx} + g_{xy}}. \tag{15}$$

To specify the boundary conditions the bulk action $S_{f\,\text{bulk}}$ in (13) needs to be supplemented by a boundary action $S_{f\,\text{bnd}}$. This action encapsulates the choice of standard or alternative quantization and any additional sources on the boundary. We adopt the usual maximally symmetric boundary action of [55, 56] but add the coupling to an external source $\mathcal{D}$ (which may break the Lorentz invariance or some other symmetry):

$$S_{f\,\text{bnd}} = \oint_{\tilde{z}=1-\epsilon} d^3x\sqrt{-h}\left[s\frac{i}{2}\bar{\Psi}_+(t,x,y,\tilde{z})\Psi_+(t,x,y,\tilde{z}) + \bar{\Psi}_-(t,x,y,\tilde{z})i\mathcal{D}\Psi_-(t,x,y,\tilde{z})\right],$$

$$s = -1, \quad \Psi_\pm = \frac{1}{2}(1 \pm \Gamma^z)\Psi, \tag{16}$$

where $\epsilon \to 0$ is the UV cutoff (computing the action at distance $\epsilon$ away from the boundary) and $\mathcal{D}$ is any sufficiently well-behaving function which can be interpreted as the external source of $\Psi$ (in the simplest interpretation it is a kinetic term on the boundary and contains derivatives with respect to $t, x, y$). The sign in front of the first term, denoted by $s = \mp 1$, corresponds to standard/alternative quantization [57]. For now we choose the sign $s = -1$, for the standard quantization, but eventually we will invert the propagator to work in the alternative one. This is equivalent to having a negative fermion mass $-1/2 < m \le 0$, so we have the conformal dimension given by the usual formula:

$$\Delta = \frac{3}{2} + m. \tag{17}$$

We discuss the choice and physical meaning of $\mathcal{D}$ in subsection 3.3; we will first finish the derivation of the expression for the field theory propagator.

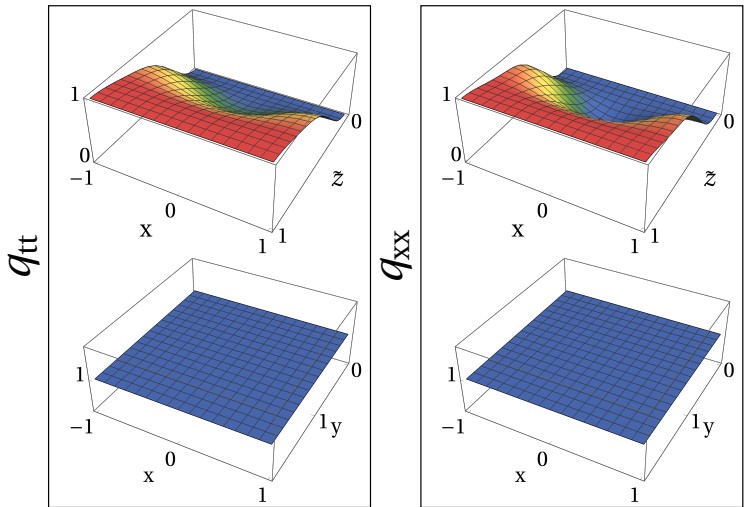

Figure 3: Numerically computed metric components $q_{tt}$ (left) and $q_{xx}$ (right), along an $x-\tilde{z}$ slice at $y = 0.5$ (top) and along an $x-y$ slice at the AdS boundary $\tilde{z} = 1$ (bottom), for the same parameter values as the previous two figures ($(\alpha, \delta) = (-1.5, 1)$, $T = 0.1$, $\mu_0 = 1$, $\delta\mu/\mu_0 = 1$, $Q = 1/2$). The top panels show that the metric becomes homogeneous both for $\tilde{z} = 0$ (there is no lattice in IR) and for $\tilde{z} = 1$ (the UV asymptotics is pure AdS). In the bottom row we show explicitly that the functions $q_{tt}(x, y; \tilde{z} = 1), q_{xx}(x, y; \tilde{z} = 1)$ are flat and equal unity, which provides a sanity check on the numerics. The oscillatory nature of the solution is thus only seen at intermediate $\tilde{z}$ scales.

As usual, we can Fourier-transform the wavefunction from time $t$ to frequency $\omega$. On the other hand, the momentum in the periodic lattice background is only defined up to a Brillouin zone, as determined by the Bloch theorem [9–11]. Therefore, instead of a Fourier transform to the plane wave basis we expand the wavefunction over the Bloch states (plane waves modulated by periodic functions). The periodicity of the Bloch states is fixed by the umklapp vector $K$, so we have the following expansion for $\Psi$:

$$
\begin{aligned}
\Psi(t, x, y, \tilde{z}) &= \int \frac{d\omega}{2\pi} \int \frac{d^2\mathbf{k}}{(2\pi)^2} \psi_{\omega\mathbf{k}}(x, y, \tilde{z}) e^{-i\omega t + ik_x x + ik_y y} \\
&= \int \frac{d\omega}{2\pi} \int \frac{d^2\mathbf{k}}{(2\pi)^2} \sum_{n_x=-\infty}^{\infty} \sum_{n_y=-\infty}^{\infty} \psi_{\omega\mathbf{k}}^{(n_x, n_y)}(\tilde{z}) e^{-i\omega t + i(k_x + n_x K)x + i(k_y + n_y K)y},
\end{aligned} \quad (18)
$$

where the Bloch momentum $\mathbf{k} = (k_x, k_y)$ is in the first Brillouin zone: $k_{x,y} \in (-K/2, K/2)$, $n_{x,y}$ are the numbers of the Brillouin zones, and the $\pm$ components of the bispinor are defined analogously to $\Psi_\pm$ in (16):

$$
\begin{aligned}
\psi_{\omega\mathbf{k}}^{(n_x, n_y)}(\tilde{z}) &= \begin{pmatrix} \psi_{\omega\mathbf{k}+}^{(n_x, n_y)}(\tilde{z}) \\ \psi_{\omega\mathbf{k}-}^{(n_x, n_y)}(\tilde{z}) \end{pmatrix}, \\
\psi_{\omega\mathbf{k}}(x, y, \tilde{z}) &= \begin{pmatrix} \psi_{\omega\mathbf{k}+}(x, y, \tilde{z}) \\ \psi_{\omega\mathbf{k}-}(x, y, \tilde{z}) \end{pmatrix} = \sum_{n_x=-\infty}^{\infty} \sum_{n_y=-\infty}^{\infty} \begin{pmatrix} \psi_{\omega\mathbf{k}+}^{(n_x, n_y)}(\tilde{z}) \\ \psi_{\omega\mathbf{k}-}^{(n_x, n_y)}(\tilde{z}) \end{pmatrix} e^{in_x Kx + in_y Ky}.
\end{aligned} \quad (19)
$$

The period $K$ is related to $Q$ as $K = 2\pi Q$. As we see, the wavefunction (and thus the Green function) on the lattice has an infinity of contributions from different Brillouin zones $n_x, n_y$. We now have all the ingredients for the Dirac equation. At this point we adopt the following

representation of gamma matrices:

$$\Gamma^{\mathrm{t}} = \begin{pmatrix} i\sigma^1 & 0 \\ 0 & i\sigma^1 \end{pmatrix}, \quad \Gamma^{\mathrm{x}} = \begin{pmatrix} -\sigma^2 & 0 \\ 0 & \sigma^2 \end{pmatrix}, \quad \Gamma^{\mathrm{y}} = \begin{pmatrix} 0 & \sigma^2 \\ \sigma^2 & 0 \end{pmatrix}, \quad \Gamma^{\mathrm{z}} = \begin{pmatrix} -\sigma^3 & 0 \\ 0 & -\sigma^3 \end{pmatrix}, \quad (20)$$

we write the two spinors $\psi_{\omega\mathbf{k}\pm}$ as

$$\psi_{\omega\mathbf{k}\pm}(x, y, \tilde{z}) = \begin{pmatrix} a_{\omega\mathbf{k}\pm}(x, y, \tilde{z}) \\ b_{\omega\mathbf{k}\pm}(x, y, \tilde{z}) \end{pmatrix}, \quad (21)$$

and obtain

$$\left(\partial_{\tilde{z}} + \Omega_{\tilde{z}} + \frac{2m\tilde{z}}{1-\tilde{z}^2}\sqrt{\frac{q_{zz}}{f}}\right) a_{\omega\mathbf{k}\pm} + \left(-\Omega_t \pm \Omega_x - \kappa\mathcal{K}\partial_{\tilde{z}} A_t\right) b_{\omega\mathbf{k}\pm} - \Omega_y b_{\omega\mathbf{k}\mp} = 0,$$

$$\left(\partial_{\tilde{z}} - \Omega_{\tilde{z}} - \frac{2m\tilde{z}}{1-\tilde{z}^2}\sqrt{\frac{q_{zz}}{f}}\right) b_{\omega\mathbf{k}\pm} + \left(\Omega_t \pm \Omega_x - \kappa\mathcal{K}\partial_{\tilde{z}} A_t\right) a_{\omega\mathbf{k}\pm} - \Omega_y a_{\omega\mathbf{k}\mp} = 0, \quad (22)$$

$$\Omega_t = \Omega_t\left(x, y, \tilde{z}; \omega\right), \quad \Omega_{x,y} = \Omega_{x,y}\left(x, y, \tilde{z}; \mathbf{k}\right), \quad \Omega_{\tilde{z}} = \Omega_{\tilde{z}}\left(x, y, \tilde{z}; \mathbf{k}\right), \quad \mathcal{K} = \mathcal{K}\left(x, y, \tilde{z}\right).$$

We again refrain from writing out in full the (very cumbersome) expressions for the $\Omega$ and $\mathcal{K}$ terms which contain also the spin connection; they would not contribute to physical understanding anyway. The result is a linear (partial) differential equation, and thus presents no problems with numerical convergence. At the end of Appendix B we discuss some practicalities concerning the numerical implementation and solution of (22).

So far we have not discussed the dipole coupling term. As we know, the dipole couples to the field strength $F_{\mu\nu}$. This term was proposed in [44] and further explored in [14,45,58–61] and other works. Its action is to simulate a gap in the spectrum. Roughly speaking, it does so by shifting the effective momentum vector $\mathbf{k}$ so that, depending on the dipole strength $\kappa$ and the position of the Fermi momentum $k_F$, the quasiparticle is "skipped" and we only have low-intensity incoherent background in some interval of momenta. More elaborate explanations are found in [45,61]: the authors show that the gap arises from the zero-pole duality between the Green functions in standard and alternative quantization. Since the dipole term mixes the leading and subleading mode at the boundary (see later in Eq. (24)), a pole can appear in the denominator of the propagator, i.e. in the self-energy, producing zeros in the spectrum. This is quite in line with the real-world Mott physics where likewise the self-energy diverges, however in order to really understand the relation to Mottness[6] one should certainly put the system on the lattice. We have done so, and we will comment both on the phenomenology of nonzero $\kappa$ systems and on possible relations to the Mott phase of the Hubbard model.

## 3.2 Boundary conditions and the Green function

It is well known how to obtain the retarded real-time Green function on the CFT side from the bulk solutions with infalling boundary condition on the horizon.[7] Near-horizon expansion of the equation (22) in the background (7) in the Bloch wave representation (18) yields schematically:[8]

$$\begin{pmatrix} \psi_{\omega\mathbf{k}+}(x, y, r \to r_h) \\ \psi_{\omega\mathbf{k}-}(x, y, r \to r_h) \end{pmatrix} = (r - r_h)^{-\frac{i\omega}{2\pi T}} \left[ \begin{pmatrix} \psi_{\omega\mathbf{k}+}^{(0)}(x, y) \\ -i\psi_{\omega\mathbf{k}+}^{(0)}(x, y) \end{pmatrix} + \begin{pmatrix} \psi_{\omega\mathbf{k}+}^{(1)}(x, y) \\ i\psi_{\omega\mathbf{k}+}^{(1)}(x, y) \end{pmatrix} (r - r_h) + \dots \right]. \quad (23)$$

---

[6]There are different views and definitions on what exactly defines Mottness. We understand it mainly as the presence of a gap due to "traffic jam" from the interactions, which is seen as the shift of the spectral weight from high to low frequencies when the doping changes.

[7]Since we are always at finite temperature, the notion of infalling solution is unambiguous. For the analytical near-horizon expansion we use the $r$ coordinate instead of $\tilde{z}$.

[8]Although we have earlier denoted the number of the Brillouin zone by upper indices and now we use the upper indices to count the order of the terms in near-horizon expansion, we believe this will not cause confusion: previously we had a pair of numbers $(n_x, n_y)$ and now just a single number; besides, we have already emphasized that solely the case $n_x = n_y = 0$ is considered throughout the paper.

Therefore, we have one independent spinor at the horizon at both leading ($\Psi_+^{(0)}$) and subleading ($\Psi_+^{(1)}$) order, as it should be. The coefficients $\psi_{\omega\mathbf{k}+}^{(0)}, \psi_{\omega\mathbf{k}+}^{(1)}$ can be determined from the near-horizon expansion of the Dirac equation, providing the Dirichlet boundary condition in the IR. We discuss the IR limit of the Dirac equation in more detail in the next section in order to learn about the scaling of the spectral function and self-energy. For now it suffices to say that (23) provides the IR boundary conditions.

Near the AdS boundary the behavior is universal (for the sake of brevity we do not write explicitly the $\omega$ and $\mathbf{k}$ dependence of the UV coefficients $A_\pm, B_\pm$ and the propagators):

$$\begin{pmatrix} \psi_{\omega\mathbf{k}+}(x,y,\tilde{z}\to 1) \\ \psi_{\omega\mathbf{k}-}(x,y,\tilde{z}\to 1) \end{pmatrix} = \begin{pmatrix} B_+(x,y) \\ 0 \end{pmatrix} \left(1-\tilde{z}^2\right)^{3/2+m} + \begin{pmatrix} 0 \\ A_-(x,y) \end{pmatrix} \left(1-\tilde{z}^2\right)^{3/2-m} + \dots \quad (24)$$

Inserting the near-boundary expansion into the boundary action (16)[9] we get

$$S_{f\,\mathrm{bnd}} \sim \epsilon^{-3}\left[\frac{1}{2}\bar{B}_+(x,y)A_-(x,y) + \bar{A}_-(x,y)i\mathcal{D}A_-(x,y)\right]. \quad (25)$$

It remains to determine the Green function from this action.

### 3.2.1 The Green function and its regularization

The holographic dictionary [1] and more specifically the prescriptions of [20, 23, 43, 55, 56], define the Green function in the following way. Since $A_-$ and $B_+$ are both finite at the boundary for $|m| < 1/2$ (the $m$ values that we compute), there are two possible choices for $G_R$ in field theory, known as the standard and alternative quantization. In the standard quantization, the propagator in absence of the source $\mathcal{D}$ is defined by the ratio of the subleading and the leading branch. In the alternative quantization, the propagator is the inverse of the standard-quantization propagator, i.e. the ratio of the leading and the subleading branch. More precisely, we have:[10]

$$G_R^{\mathrm{std}}|_{\mathcal{D}=0} = i\epsilon^{-2m}B_+\sigma^1 A_-^{-1}, \quad G_R^{\mathrm{alt}}|_{\mathcal{D}=0} = \left(G_R^{\mathrm{std}}|_{\mathcal{D}=0}\right)^{-1} = -i\epsilon^{2m}A_-\sigma^1 B_+^{-1}. \quad (26)$$

In the presence of the source, the above expression for the standard quantization generalizes in a straightforward way, taking the second variation of the total action (25) with respect to $A_-$:

$$G_R^{\mathrm{std}} = i\epsilon^{-2m}B_+\sigma^1 A_-^{-1} + i\epsilon^{-1}\mathcal{D} = G_R^{\mathrm{std}}|_{\mathcal{D}=0} + i\epsilon^{-1}\mathcal{D}. \quad (27)$$

In order to get the Green function in the alternative quantization we keep the relation $G_R^{\mathrm{alt}} = \left(G_R^{\mathrm{std}}\right)^{-1}$ also in the presence of the source, getting from (27):

$$G_R^{\mathrm{alt}} = \left(\left(G_R^{\mathrm{alt}}|_{\mathcal{D}=0}\right)^{-1} + i\epsilon^{-1}\mathcal{D}\right)^{-1}. \quad (28)$$

This prescription is also in accordance with the regularization in [43] and its interpretation in [20]. The term $\epsilon^{-2m}$ in (26) is merely the dimensional regulator coming from the scaling of the boundary fermionic operator, and the $\epsilon^{-1}$ term in front of $\mathcal{D}$ corresponds to the canonical dimension of the operator $\mathcal{D}$. Here we see the role of the boundary operator $\mathcal{D}$ in the expression for the Green function. Such a nontrivial "source" was first proposed already in [56] and used in [43] to regulate the $\omega$-dependence in order to satisfy the ARPES sum rule. We use it for precisely the same purpose, though we will motivate it in a slightly different way in the

---

[9]The bulk action vanishes on-shell as it is proportional to the Dirac equation.

[10]We write $G_R$, for the retarded Green function, as this is the one that we compute in this paper; but in fact the expressions for the Green function in terms of boundary values of the bulk fermion are independent of the contour prescription.

next subsection. From now on we will write just $G_R$ instead of $G_R^{\text{alt}}$ as we only calculate the alternative quantization.

The last stroke is to implement the above procedure for an inhomogeneous system, where the wavefunction depends on $x, y$ as in (24). In principle, everything could be done just like in Eqs. (26-28), and the result would be a local Green function (depending on the coordinates $x, y$). But that is not appropriate for a photoemission "experiment": ARPES uses photons in plane wave states, thus we want the Fourier transform of the local propagator to the momentum space. This procedure is discussed in [7, 9–11]. We start from the expansion of $\psi_{\omega\mathbf{k}\pm}(x, y, \tilde{z})$ in terms of $\psi_{\omega\mathbf{k}\pm}^{(n_x, n_y)}(\tilde{z})$ and consequently the expansion of $a_{\omega\mathbf{k}\pm}, b_{\omega\mathbf{k}\pm}$ as

$$\begin{pmatrix} a_{\omega\mathbf{k}\pm}(x, y, \tilde{z}) \\ b_{\omega\mathbf{k}\pm}(x, y, \tilde{z}) \end{pmatrix} = \sum_{n_x=-\mathcal{N}}^{\mathcal{N}} \sum_{n_y=-\mathcal{N}}^{\mathcal{N}} \begin{pmatrix} a_{\omega\mathbf{k}\pm}^{(n_x, n_y)}(\tilde{z}) \\ b_{\omega\mathbf{k}\pm}^{(n_x, n_y)}(\tilde{z}) \end{pmatrix} e^{in_x K_x + in_y K_y}, \tag{29}$$

where now $a_{\mathbf{k}\pm}^{(n_x, n_y)}$ and $b_{\mathbf{k}\pm}^{(n_x, n_y)}$ only depend on $\tilde{z}$, with $\omega$ and $\mathbf{k}$ having the role of parameters, so these functions satisfy ordinary differential equations, obtained by plugging (29) into (22). We have now truncated the formally infinite expansion in $(n_x, n_y)$ by a finite zone number $\mathcal{N}$. The next step is to expand the UV coefficients $A_\pm, B_\pm$ from (24) analogously to the expansion (29) and to insert them into the Green function in the alternative quantization from (26), getting:[11]

$$A_\sigma^{(n_x, n_y)}(\omega, \mathbf{k}) = i\epsilon^{-2m} \sum_{\sigma'=\pm} \sum_{n_x=-\mathcal{N}}^{\mathcal{N}} \sum_{n_y=-\mathcal{N}}^{\mathcal{N}} G_{n_x, n_y}^{n'_x, n'_y}(\omega, \mathbf{k}) B_{\sigma'}^{(n'_x, n'_y)}(\omega, \mathbf{k}). \tag{30}$$

The Green function thus becomes a matrix with both spinor and Brillouin zone indices. Computing the full matrix for a 2D lattice would be a huge computational job. But the off-diagonal terms (with $n'_x \neq n_x, n'_y \neq n_y$) are usually suppressed, at least for the weak lattice regime defined in Eq. (6); for strong lattices this approximation is of questionable validity. Since we stay in the weak lattice regime, we follow all previous works on ARPES spectra on lattices [7, 9–12] and disregard all off-diagonal terms, putting $(n'_x, n'_y) = (n_x, n_y)$ in (30) and eliminating the sum over the zone numbers.[12] One might worry that this is a basis-dependent notion but, as pointed out in [9], taking the trace when computing the spectral function eliminates the basis dependence as the trace is an invariant. The source is a plane wave, as in real ARPES experiments, therefore $A_-^{(n_x, n_y)} = 0$ for any $n_x \neq 0$ or $n_y \neq 0$ is the boundary condition for the Dirac equation (the normalization is arbitrary). We then extract the components $B_\pm^{(n_x, n_y)}$ with the cutoff $\mathcal{N} = 2$ and sum them. This yields directly the values of the diagonal terms in $G_R$. Once $G_R$ is computed in this approximation, the spectral function is obtained as the trace of its imaginary part: $A(\omega, \mathbf{k}) = -\text{Im Tr } G_R(\omega, \mathbf{k})/\pi$.

We are now ready to state the algorithm for computing the holographic Green function:

1. For chosen parameters (scaling exponents) $(\alpha, \delta)$, calculate the background (metric, gauge field, dilaton) by solving numerically the EMD system of equations defined by the action (1).

2. With this background as the input, calculate the wavefunction of the probe fermion by solving numerically the Dirac equation defined by the bulk action (13) and the boundary action (16).

---

[11]For the wavefunctions we have written the $\omega$ and $\mathbf{k}$ dependence in the subscript, as the wavefunctions are really functions of coordinates. But for the coefficients of the UV expansion, which do not depend on any coordinate, it is natural to write $\omega, \mathbf{k}$ as function arguments.

[12]Actually, since earlier works on ARPES spectra considered unidirectional lattices, they only had to deal with a single $n$.

3. From this wavefunction (specifically its behavior at $\tilde{z} \to 1$) as the input, determine the Green function $G_R$ as given in (28).

## 3.3 A detour – boundary sources and their holographic interpretation

A holographic action may incorporate some boundary source $\mathcal{D}$ as long as it does not violate the action principle and the AdS/CFT dictionary. Such a term is usually interpreted in the semiholographic sense, i.e. as some (typically weakly interacting or free) theory coupled to the strongly interacting holographic system. In [20,43] the ARPES sum rule was enforced by coupling the holographic fermion to a free fermion of zero mass (with $\mathcal{D}$ being just the kinetic term). We could employ exactly the same recipe to arrive at normalizable spectral functions, however we prefer to choose a slightly different route for two reasons (1) to facilitate the comparison to the Hubbard model later in the paper, we want a faster decay at large $\omega$ (2) it is handy to have the source term which at least in principle can itself be obtained from holography. In that case, the source $\mathcal{D}$ is effectively another holographic system (in a different, "auxiliary" AdS space) coupled to the main model only through the boundary coupling.

This subsection is something of a detour – it addresses a technicality which is mainly important for the comparison to the Hubbard model and otherwise not really crucial; it defines a cutoff which (being just a cutoff) could also be introduced by hand; and it is more technical and less related to condensed matter applications than most of the paper. It can thus be skipped on first reading. Now we give a very quick derivation of the cutoff (very quick because the steps and the model itself are long known and we merely employ them in a novel context).

We propose to employ a soft wall model which introduces a dilaton-like scalar $\varphi$ coupled to the auxiliary fermion $\chi$. Such models have been widely studied in the context of AdS/QCD. In analogy with [62] (although there are many other papers with similar models), we start from the action[13]

$$S_{\mathcal{D}} = i \int d^4 x \sqrt{-g}\, \bar{\chi} \left[ \frac{1}{2}\left( \overleftarrow{\not{D}} + \overrightarrow{\not{D}} \right) - M - \Gamma^0 \Gamma^1 \Gamma^2 \varphi \right] \chi \,. \tag{31}$$

The dilaton $\varphi$ and the fermion $\chi$ are both considered in the probe limit, so the metric of the auxiliary system remains pure AdS. Following [62], we adopt the quadratic dilaton profile $\varphi = \varphi_0 z^2$; we use $z$ as the radial coordinate so the boundary is at $z = 0$ and the interior is at $z$ large. The Dirac equation for $\chi$, written as $\chi = (x_+, x_-, y_+, y_-)$ is now

$$\left( \partial_z \pm \frac{M + \varphi_0 z^2}{z} \right) x_\pm \mp (\omega \mp k_x) x_\mp = 0 \,, \tag{32}$$

and the equation for $y_\pm$ is obtained from the above one by putting $(k_x \mapsto -k_x, \varphi_0 \to -\varphi_0)$. This can be solved analytically in terms of the Tricomi confluent hypergeometric functions $U(a, b, x)$:

$$\chi_\pm(\omega, k, z) = e^{\pm \varphi_0 z^2/2} z^{\pm M} U\left( \frac{\omega^2 - k^2}{4\varphi_0}, \frac{1}{2} \pm M, \mp \varphi_0 z^2 \right) \,. \tag{33}$$

Following the usual recipe – expanding the solutions near $z = 0$ and taking the ratio of the two independent branches of the solution, the propagator of the auxiliary fermion is

$$G_\chi = \mathcal{D} = \frac{\Gamma\left( \frac{\omega^2 - k^2}{4\varphi_0} \right)}{\Gamma\left( \frac{\omega^2 - k^2}{4\varphi_0} + \frac{1}{2} - M \right)} \,, \tag{34}$$

---

[13]The reader might protest that this form of coupling to the dilaton violates the Lorentz invariance. This is true but it is unavoidable: a cutoff in energy only cannot be Lorentz-invariant. On the other hand, a cutoff in both energy and momenta would not be justified, as the momentum is automatically regulated on the lattice. We have employed a similar regulator in a very different context (quantum electron stars) in [63].

with $\Gamma$ being the usual gamma function. Notice that there is no chemical potential and of course no lattice in auxiliary AdS (the latter fact allows us to assume $k_y = 0$). The function $\mathcal{D}$ has a fast growth with $\omega$ for large and positive $M$. Inserting it into Eq. (28), we see that this in turn means a fast decay of the "physical" (as opposed to auxiliary) Green function $G_R$. Essentially, $\mathcal{D}$ acts like a regulator, with little influence at low energies but rapidly suppressing the spectral weight for $\omega$ large.

Of course, one can also adopt a pragmatic viewpoint and simply introduce a regulator by hand. For the purposes of this paper this is equally valid, however the viewpoint of this subsection may be of interest on its own; it is essentially the implementation of the idea from [64] to couple multiple AdS spaces in order to obtain a model with several different scales.

## 4 Fermionic spectral functions

We come now to the core of the paper, analyzing the spectral functions of our system as a function of frequency. From now on all dimensionful quantities in the paper are stated in units of the wavevector $Q$ as defined in Eq. (4).

### 4.1 General phenomenology and the "phase diagram"

Let us first return to the homogeneous Fermi and non-Fermi liquids of [27]. The near-horizon analysis in that paper shows that the crucial parameter is the combination

$$\beta + \gamma = 1 + \frac{\alpha^2 - \delta^2}{4 + (\alpha + \delta)^2}, \tag{35}$$

where $\alpha$ and $\delta$ are the exponents of the dilaton potentials in (2) and $\beta, \gamma$ are the IR scaling exponents of (10). For $\beta + \gamma > 1$, the imaginary part of the self-energy $\Sigma$ (obtained by solving the near-horizon effective Schrödinger equation as in [23]) is exponentially small near the Fermi surface $\omega \to 0$ and the quasiparticle pole is stable and long-living. For $\beta + \gamma < 1$, similar analysis in [27] shows that Im$\Sigma$ is always of order unity, hence there is no sharp quasiparticle. Properties at finite frequencies and of course the lattice effects cannot be described by this simple reasoning, but the above serves to show that we should expect at least two regimes, with and without a sharp quasiparticle peak. Since the combination $\beta + \gamma$ can be varied even with one of the fundamental exponents $\alpha, \delta$ fixed, we choose $\delta = 1$ throughout the whole paper and vary $\alpha$, as well as the fermion dimension $\Delta$ (defined in (17)) which characterizes the fermionic operator in field theory (not the background). For $\delta = 1$, the critical value $\beta + \gamma = 1$ is reached for $\alpha = -1$:[14]

$$\delta = 1 \Rightarrow \beta + \gamma = 1 + \frac{\alpha^2 - 1}{4 + (\alpha + 1)^2}, \tag{36}$$

so we will use the notation $\beta + \gamma > 1$ and $\alpha < -1$ interchangeably. Therefore, all our "phase diagrams" will be $\alpha - \Delta$ diagrams, keeping the fixed value of $\delta$. We put "phase diagram" under quotation marks as we have not checked the free energy and there is no reason to believe that any sharp phase transition happens (indeed, $\Delta$ cannot influence thermodynamic phases at all as it is a probe parameter).

From now on we work on the lattice with $\delta\mu/\mu_0 = 0.5$ unless specified otherwise and express all quantities in units of $\mu_0$. Numerical results (soon to be shown in detail) show two key phenomena (at least for some $(\alpha, \Delta)$) values: quasiparticle peaks and broad bumps at

---

[14]Remember that $\alpha < 0$.

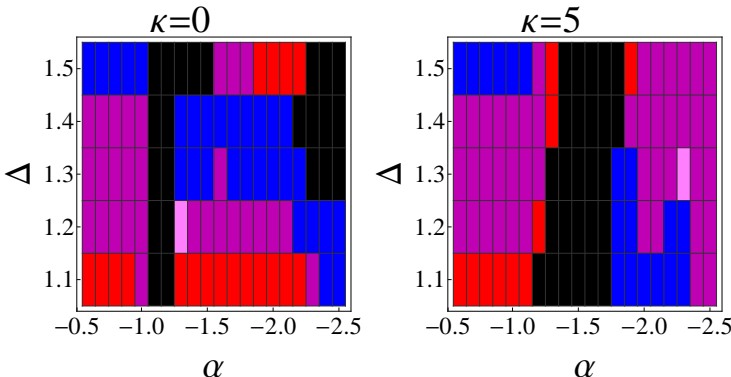

Figure 4: Presence of a coherent quasiparticle (QP) or a gap between the central peak and the Hubbard bands, as a function of the background exponent $\alpha$ and the conformal dimension of the probe fermion $\Delta$, for no dipole coupling (left) and for strong dipole coupling (right): black – no QP and no gap, blue – QP only, red – gap only, violet – QP and the gap. Speaking very roughly, the violet regions correspond to Mott-like and Hubbard-like physics, where a more or less coherent central QP peak is surrounded by but well-separated from intermediate-frequency bands. The clear and deep local minima of the fit to quantum Monte Carlo solutions of the Hubbard lattice, denoted by light pink fields, are both located within the violet (QP+bands) regions. We will refer to this last feature again in Section 5 when discussing the Hubbard fit.

finite energy (to the left and to the right from $\omega = 0$). These finite-frequency maxima are separated from the central maximum around $\omega = 0$ by soft gaps; we call them (Hubbard-like) bands but one should bear in mind that the actual physics might be very different from the bands in the Hubbard Hamiltonian (running a bit forward, it still has to do with umklapp but in detail it is certainly different from Hubbard bands). The key property of the bands is precisely the soft gap (usually the spectral weight does not drop exactly to zero) separating them from the low-frequency part. In Fig. 4 we map the areas with a quasiparticle and/or a gap between the central peak and at least one of the bands (upper or lower). The central peak may be a quasiparticle or just a broad maximum, and the criterion for the gap is that there is a point (or an interval) at finite $\omega$ where the spectral weight $A(\omega, \mathbf{k})$ is exponentially low compared to the ratio of the peak maximum to the "background" value of the spectral function. For a Fermi liquid, this ratio grows roughly as $T^2$ so this essentially means that the gap has to be exponentially suppressed in inverse temperature (squared); for a non-Fermi liquid, it is harder to estimate the temperature scaling but it still makes sense to require that the gap be exponentially small in peak height.[15]

The general structure of different "phases" is seen in Fig. 5, containing the numerical EDCs for three backgrounds ($\alpha = -0.6, -1.1, -2.5$), each for a range of conformal dimensions and for now with $\kappa = 0$. As we have predicted, the gaped spectrum for $\alpha = -0.6$ (left panel) develops a sharp quasiparticle peak for increasing $\Delta$, whereas the gapless spectrum of $\alpha = -2.5$ loses a coherent quasiparticle as $\Delta$ grows (right panel). In the central panel we show the case

---

[15]At this place we should comment on how we differentiate between quasiparticles and finite-width peaks ("bumps"). The ultimate criterion (in absence of good analytic insight) is to compute the spectral function, in the bottom half of the complex $\omega$ plane – then one can directly see the pole and differentiate it from a branch cut. However, such a calculation was too demanding for our present work (it requires a large number of points and the convergence becomes progressively worse as we increase the magnitude of the imaginary part of $\omega$). Therefore, we have simply scanned the peak in real $\omega$ with increasing resolution, and decided we have the quasiparticle if the peak width is at most $O(1)$ times the temperature $T$. This is not quite satisfying but is apparently the only choice in absence of firm analytic results.

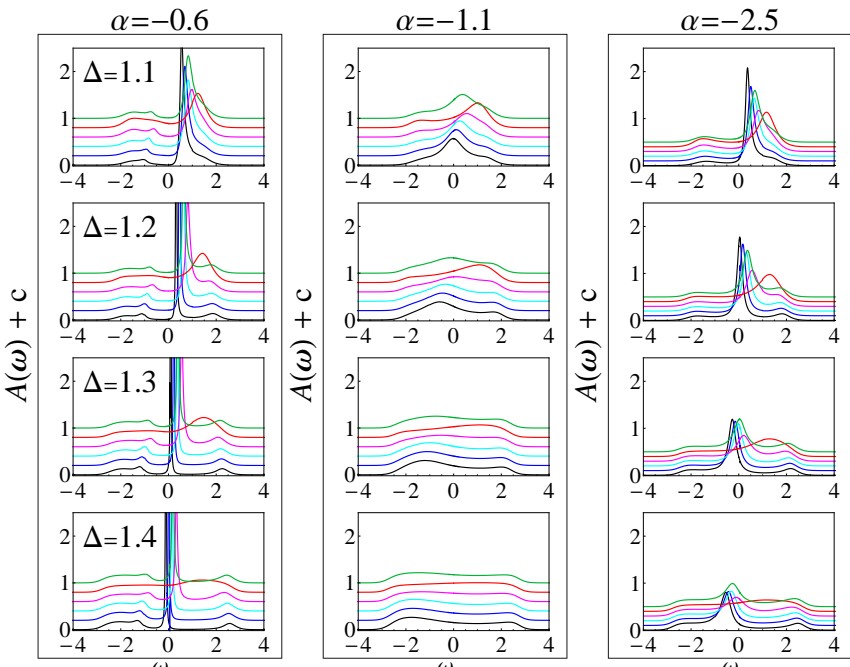

Figure 5: Energy spectra for six standard momentum values, taken along the high-symmetry path ΓXMΓ ((0,0) black, ($\pi/2$,0) blue, ($\pi$,0) cyan, ($\pi$,$\pi/2$) magenta, ($\pi$,$\pi$) red, ($\pi/2$,$\pi/2$) green), for three different backgrounds with $\alpha = -0.6, -1.1, -2.5$, each for four conformal dimensions. In the first case, the sharp peak and the gap between the peak and the bands are both present; in the second case, the spectrum only has a broad and featureless maximum, and in the third there is again a quasiparticle but without a clear gap separating it from the bands. We shift the spectral curves by a constant $c$ for each subsequent momentum value for easier viewing.

with a featureless spectrum of $\alpha = -1.1$. Notice that the peak is in general asymmetric so the quasiparticle is not of Fermi liquid type.[16] In this and subsequent figures we give the EDCs for six momenta placed along the ΓXMΓ (high-symmetry) path for the square lattice: $(0,0)$, $(\pi/2,0)$, $(\pi,0)$, $(\pi,\pi/2)$, $(\pi,\pi)$ and $(\pi/2,\pi/2)$; we call this the standard momentum set.

Temperature dependence of the EDCs is shown in Fig. 6 for the standard set of momenta and in Fig. 7 in the full $\omega - \mathbf{k}$ plane. The quasiparticle melts away upon heating the system, but we know that just had to happen. More interesting is the fact that the bands (soft bumps away from $\omega = 0$) also become almost indistinguishable for $T = 10$. This scale is higher than in the Hubbard model, where $T \sim 1$ is already a high-temperature regime, whereas here such regime only kicks in an order of magnitude later. We will have more to say on this matter when directly comparing to the Hubbard model (EDC actually converges toward a Gaussian at high $T$). In conclusion, the generic square lattice hyperscaling-violating model shows some essential features of Hubbard-Mott physics *in its single-particle spectra* (we do not know yet about transport and multi-particle operators): non-Fermi liquid (anomalously-

---

[16]In general, it is the asymmetry of momentum distribution curves (MDCs) that provides a litmus test for non-Fermi liquids, not the asymmetry of EDCs. However, it is known [26,27] that in the context of EMD models of our type Fermi liquid behavior always goes hand-in-hand with symmetric EDCs. In fact, this is known to be true in absence of lattice, but our weak binding analysis in subsection 4.2 extends this also to weak lattices. In Appendix D we show examples of MDCs which are indeed asymmetric, providing stronger evidence for the existence of the non-Fermi liquid phase. Finally, the asymmetry of the EDC peak only makes sense sufficiently near the Fermi surface; further away, the quasiparticle loses coherence anyway and we can say very little about its expected shape.

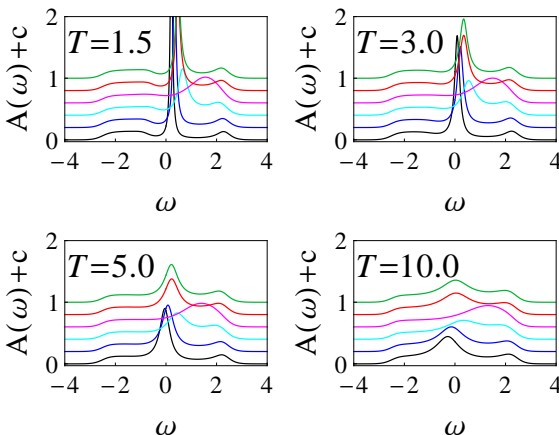

Figure 6: The EDC curves for six standard momentum values along the high-symmetry path ($(0, 0)$ black, $(\pi/2, 0)$ blue, $(\pi, 0)$ cyan, $(\pi, \pi/2)$ magenta, $(\pi, \pi)$ red, $(\pi/2, \pi/2)$ green), for a range of high temperatures, for $\mu_0 = 2.5, \alpha = -1.8, \Delta = 1.3$. As could be expected, the quasiparticle melts away and both the upper and lower band merge with the remnants of the quasiparticle. It can be shown (Section 5) that the high-temperature limit is just a single broad Gaussian peak.

scaling) quasiparticles which can be long- or short-living and Hubbard-like bands. Now we will explore the influence of dipole coupling on the system.

A robust feature seen in the first (lowest $T$) panel in Fig. 7, present also in single EDCs in Fig. 5 but perhaps not so obvious is that the quasiparticle changes intensity and sharpness as we move along the high-symmetry path in the momentum cell: around $(\pi, \pi)$ there is a drastic weakening of the quasiparticle. This feature is robust and stays there for different resolutions and parameters. In subsection 4.2 we will show that this is a consequence of umklapp in the self-energy and thus should be regarded as a generic feature of hyperscaling-violating systems on a lattice. It remains to see if this behavior can be related to real-world systems.

### 4.1.1 Dipole coupling and Mott-like physics

In absence of lattice the dipole coupling opens a gap at low frequencies, destroying the quasiparticle and leading to a Mott-like spectrum, as we already commented from the results of [14, 44, 45, 59–61]. But that mechanism crucially depends on shifting the effective momentum *and* on the zero-pole duality in the spectrum. While the latter remains true on the lattice, the former mechanism will likely work in a different way. Indeed, the mixing of modes from different Brillouin zones shifts the gap to *finite* frequencies – the net result being not a Mott gap around $\omega = 0$ but a gap separating the low-energy peak from higher enegies. If we already have a soft gap there (when $\kappa = 0$), the result is a rather hard gap. This generally brings the spectrum closer to the Hubbard-Mott physics, but in some cases, in absence of quasiparticle, the hard gap actually hinders the transfer of the spectral weight from the central peak to the bands, killing the quasiparticle (Fig. 8).

What we said above holds for lower values of the dipole coupling $\kappa$. For stronger couplings, the gap both hardens and *broadens* and for some $\alpha$ values eventually encompasses also the Fermi surface, i.e. the zero of energy. In this regime (Fig. 9) we approach a Mott insulator, eliminating the central broad peak that we had in the non-quasiparticle regime at low $\kappa$ (though the peak may still be present for some momenta, thus the system is not yet fully insulating). In Fig. 10 we show the action of $\kappa$ also in a full $(\omega, \mathbf{k})$ density plot of the spectrum – we see both the hardening of the gap and the sharpening of the central peak. Crucially, the

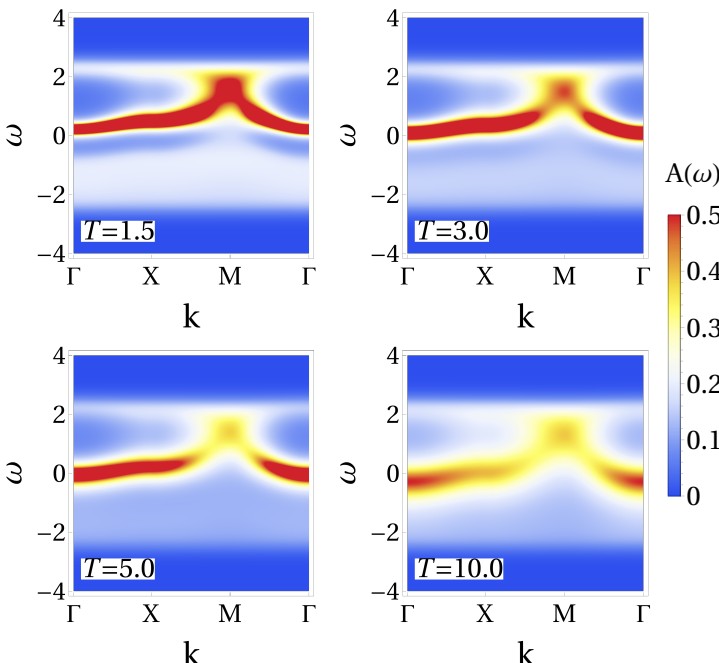

Figure 7: Same as in Fig. 6 but here with full $\omega - \mathbf{k}$ dispersion plots, to better appreciate the evolution of the spectrum with temperature.

weakening of the quasiparticle near $(\pi, \pi)$ is less prominent and the central peak is in general less momentum-dependent: this brings the system more in line with the conventional wisdom (but takes away an interesting feature which is physical and might have its place in nature).

Having explored in the main the dependence of the spectrum on proxy interactions $(\alpha, \Delta, \kappa)$, let us look how the spectrum reacts to doping (chemical potential). In Fig. 11 we increase the chemical potential, shifting the maximum of the spectral weight from $\omega < 0$ (black) to $\omega \approx 0$ corresponding to a Fermi surface of a metal (blue) to $\omega > 0$ (red), for two backgrounds and for zero or nonzero dipole coupling. This is maybe the best (although still shaky) argument toward the interpretation of $\kappa$ as a source of Mottness on the holographic lattice.[17] In absence of dipole coupling, increased doping (chemical potential) just pushes the system toward a quasiparticle regime (left panel); this may also happen for nonzero coupling (central panel) but in the appropriate place in the $(\alpha, \Delta)$ diagram we witness just a prominent lower band and a much smaller upper band with no quasiparticle, a possible signature of Mottness (right panel). Finally, we note that other bulk mechanisms (probe fermion self-interactions) have also been proposed as a source of Mottness [65]; we have not explored them in this work.

## 4.2 Perturbative analysis and matching for the holographic lattice

In this subsection we will write down the weak binding perturbation theory in the two limits (UV and IR) for the bulk Dirac equation in the hyperscaling-violating background. Essentially, we follow the approach of Hong Liu and coworkers in [23] but now in a periodic potential. A similar calculation was done in great detail in [4] for 1D probe lattice superimposed on the Reissner-Nordstrom background. We follow essentially the same logic. On one hand, our task is more difficult as the lattice is 2D and the geometry is much more complicated; on the other hand, since the lattice is a subleading (irrelevant) perturbation in the IR, the IR equations are

---

[17]In absence of lattice, as we mentioned, the situation is simpler and earlier works present a rather complete picture on how this works.

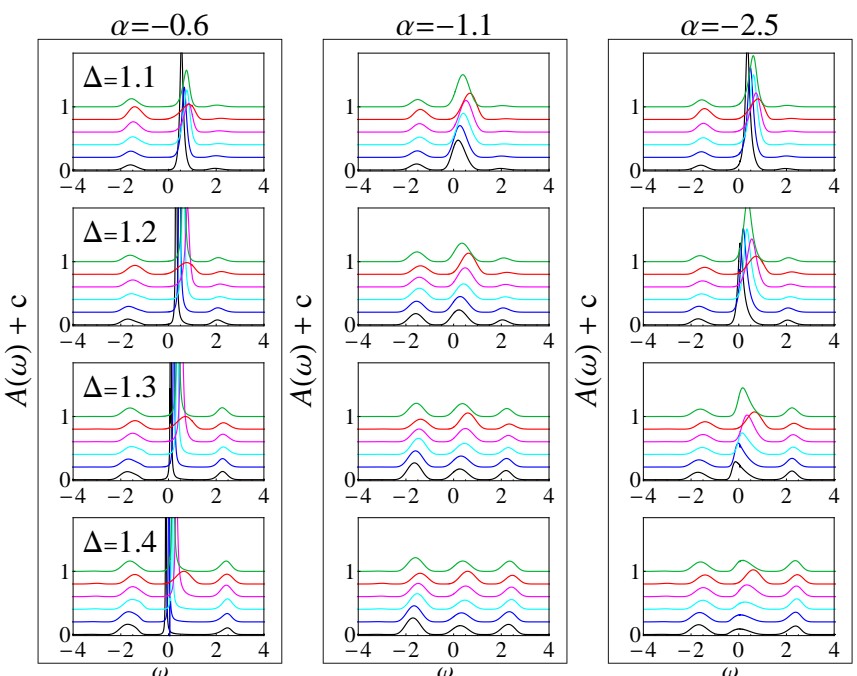

Figure 8: Energy spectra for the same (standard) momenta, scaling exponents and conformal dimensions as in Fig. 5 but with a nonzero dipole coupling $\kappa = 1.5$. The main effect of not very large $\kappa$ is to harden the gap between the central peak/quasiparticle and the bands. This is reminiscent of Hubbard and similar microscopic models, and can be explained as a combination of momentum shift introduced by $\kappa$ and the nonmonotonic $\omega$ dependence of the umklapp terms in the self-energy (see subsection 4.2).

somewhat simplified (though we still have the mixing of different Brillouin zones). We set the dipole coupling to zero in this subsection.

The basic idea is simply to expand the coefficients of the Dirac equation (22) in the $(x, y)$-dependent perturbation (proportional to $\delta\mu/\mu_0$ as a small parameter) and then to transform to the basis of Bloch states. We keep the notation from (18,19,21) with the additional convention that we write $\mathbf{n}$ for $(n_x, n_y)$. The components of the Dirac bispinor (22) are then written as

$$\psi^{(\mathbf{n})} \equiv \left( a_+^{(\mathbf{n})}, b_+^{(\mathbf{n})}, a_-^{(\mathbf{n})}, b_-^{(\mathbf{n})} \right), \tag{37}$$

so for example $a_+^{(\mathbf{n})}$ stands for $a_+(\mathbf{k} + 2\pi\mathbf{n}Q)$ with $\mathbf{k}$ from the first Brillouin zone.

Let us start from the AdS boundary and use the coordinate $r$, related to $\tilde{z}$ as in (3). From the expansion (12), we know that the metric functions $q_{\mu\nu}$ and the dilaton are constant up to second order in $1/r$ (the boundary is at $r = \infty$). Of course, the modulation of the chemical potential is relevant and present already at leading order. For a weak lattice as in Eq. (6), we can expand the coefficients of the Dirac equation both in $1/r$ (UV expansion) and in $\delta\mu/\mu_0$ (weak binding expansion); the fermionic wavefunction is likewise expanded in $\delta\mu/\mu_0$.

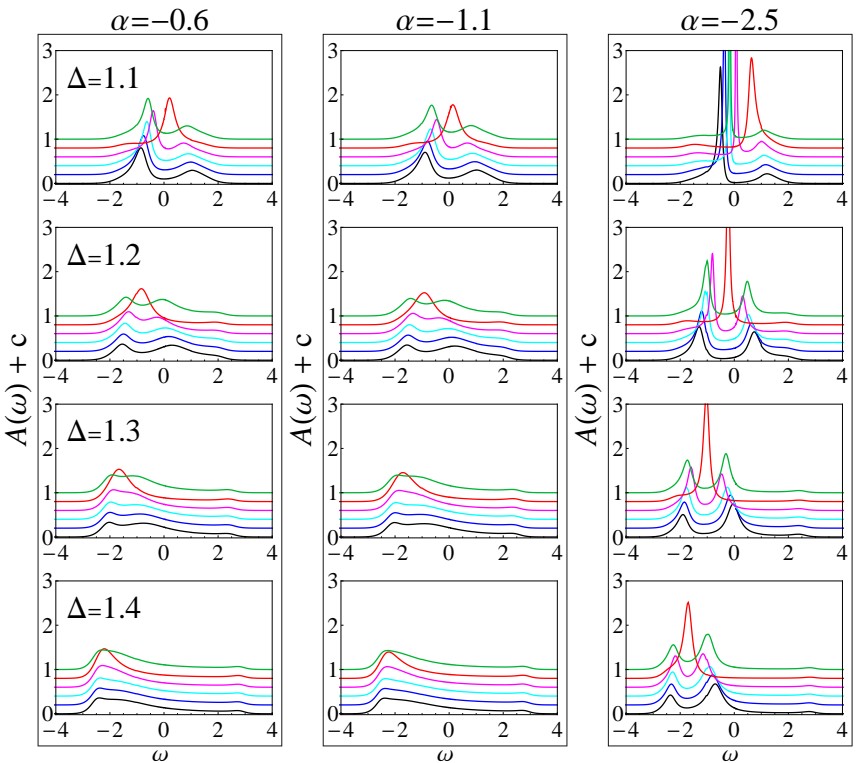

Figure 9: Energy spectra for the same (standard) momenta, scaling exponents and conformal dimensions as in Fig. 5 but with a strong dipole coupling $\kappa = 5$. Here the effect is different and more drastic than for $\kappa = 1.5$ in Fig. 8: the suppression of the spectral weight now encompasses also the Fermi surface and its vicinity (the region around $\omega = 0$), so the quasiparticle either disappears or splits into two. In this regime the dipole coupling overrides the umklapp and opens a broad gap.

The expansions for $A_t$ and $\psi$ read:

$$A_t^{(\mathbf{n})}(r) = \bar{A}_t^{(\mathbf{n})}(r) + \delta A_t^{(\mathbf{n})}(r), \tag{38}$$

$$\bar{A}_t^{(\mathbf{n})}(r) = \mu_0 \delta_{\mathbf{n}}\left(1 - \frac{1}{r} + \dots\right), \tag{39}$$

$$\delta A_t^{(\mathbf{n})} = \left(\frac{\delta\mu}{\mu_0}\right)\frac{\mu_0}{4}\delta_{|n_x|-1}\delta_{|n_y|-1}\left(1 - \frac{1}{r} + \dots\right), \tag{40}$$

$$\psi^{(\mathbf{n})}(\mathbf{k}, r) = \psi_{(0)}^{(\mathbf{n})} + \left(\frac{\delta\mu}{\mu_0}\right)\psi_{(1)}^{(\mathbf{n})} + \left(\frac{\delta\mu}{\mu_0}\right)^2 \psi_{(2)}^{(\mathbf{n})} + \dots \tag{41}$$

Of course, $\delta_{\mathbf{n}}$ stands for the product of Kronecker deltas $\delta_{n_x}\delta_{n_y}$. The UV Dirac equation at leading order in $\delta\mu/\mu_0$ includes the zero- and first-order terms, so it can be solved hierarchically by obtaining $\psi_{(0)}^{(\mathbf{n})}$ first (this is just the homogeneous solution at the boundary) and plugging it in the first-order equation to obtain $\psi_{(1)}^{(\mathbf{n})}$:

$$\left(\partial_r + \frac{m}{r}\right)a_{\pm(1)}^{(\mathbf{n})} - \frac{\omega \mp k_x + q\bar{A}_t}{r^2}b_{\pm(1)}^{(\mathbf{n})} - \frac{k_y}{r^2}b_{\mp(1)}^{(\mathbf{n})} = \frac{q\delta\mu}{4r^2}\sum_{\mathbf{n}'}b_{\pm(0)}^{(\mathbf{n}')},$$

$$\left(\partial_r - \frac{m}{r}\right)b_{\pm(1)}^{(\mathbf{n})} + \frac{\omega \pm k_x + q\bar{A}_t}{r^2}a_{\pm(1)}^{(\mathbf{n})} - \frac{k_y}{r^2}a_{\mp(1)}^{(\mathbf{n})} = \frac{q\delta\mu}{4r^2}\sum_{\mathbf{n}'}a_{\pm(0)}^{(\mathbf{n}')}, \tag{42}$$

where $\mathbf{n}' \in \{(n_x + 1, n_y + 1), (n_x + 1, n_x - 1), (n_x - 1, n_x + 1), (n_x - 1, n_y - 1)\}$.

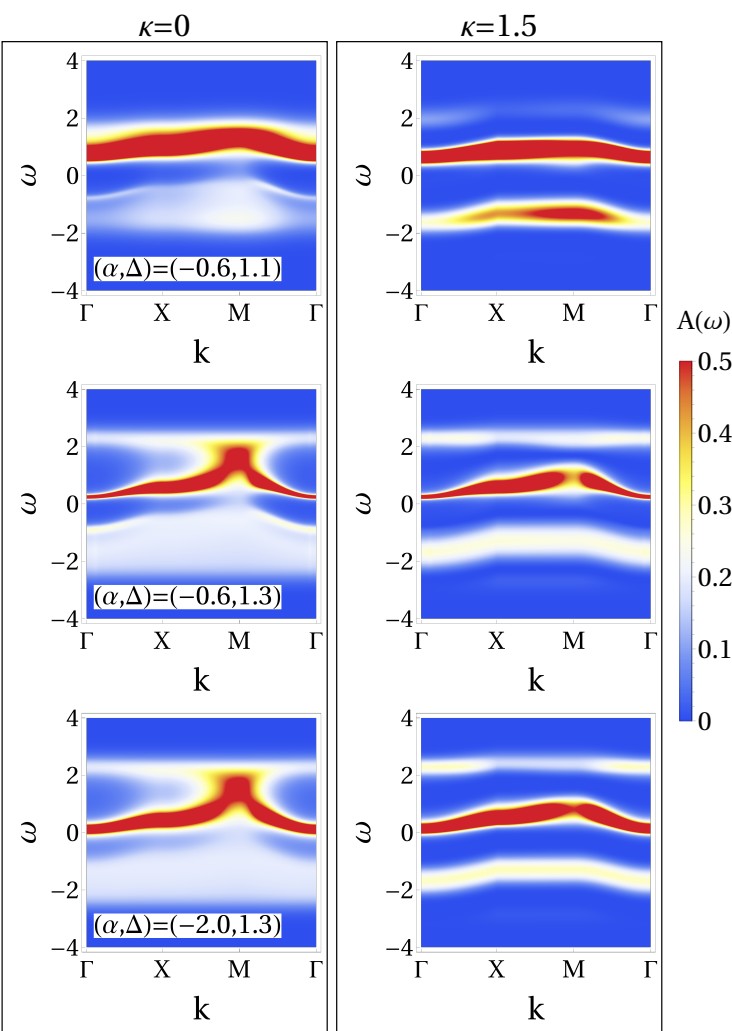

Figure 10: A set of density plots comparing the $\kappa = 0$ and $\kappa = 1.5$ spectra for a range of $(\alpha, \Delta)$ values. From first to last row the central peak evolves from a wide bump to a sharp peak, and the role of the dipole coupling (second column) is to keep the side bands well-defined and separated from the center. The weakening of the quasiparticle near the M point is less prominent with nonzero dipole coupling.

The right-hand side of the above equation, i.e. the source terms are the terms which mix different zones. After solving for $\psi_{(1)}^{(\mathbf{n})}$ we can analogously obtain the next approximation and so forth, mixing the modes from more and more zones (in Eq. (42) we have only written the first equation of the infinite hierarchy). Since the mixing of different zones only appears in the sources, the solution to any order has the same form as in Eq. (24). This was already noticed in [4] – the UV side is easy to write down.

The IR region is subtler. Despite the fact that the leading periodic corrections only appear in the second order expansion of the bulk fields, in the Dirac equation they show up at leading order already, because its coefficients mix various terms from the scaling ansatz (8) for the background. Expanding in $r - r_h$ and $\delta\mu/\mu_0$, we end up with an equation of the same form as (42), with higher order terms mixing different modes, but the coefficients and the radial dependence are more complicated. For later use it is convenient to rewrite the equation in the Schrödinger form:

$$\left(\hat{I}_{4\times 4}\partial_r^2 + \hat{M}_1\partial_r + \hat{M}_0\right)\psi_{(1)}^{(\mathbf{n})} = \mathcal{S}_{(0)}^{(\mathbf{n})}. \tag{43}$$

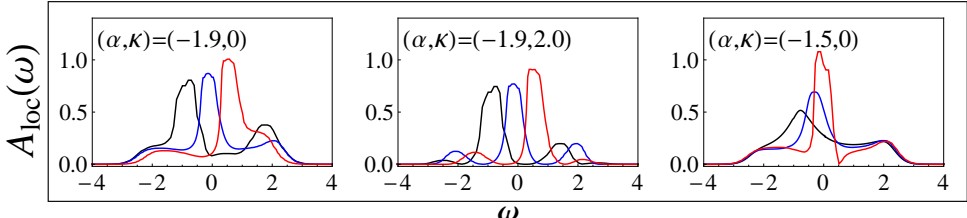

Figure 11: Local (momentum-integrated) spectral weight at frequency $\omega$ denoted by $A_{\text{loc}}(\omega)$ for three interesting cases: metallic phase without (left) and with (center) dipole coupling, for $\alpha = -1.9$, and the transition toward a Mott-insulator-like phase (right), for $\alpha = -1.5$ and no dipolar coupling. All cases are for $\Delta = 1.2$ and for three chemical potentials (dopings), in black, blue and red respectively: $\mu_0 = 0.81, 2.10, 2.70$ (left and center), $\mu_0 = 1.35, 1.70, 2.10$ (right). In the first two figures, we see the central, quasiparticle-like peak moving toward the upper band as the doping is increased; the figure with nonzero $\kappa$ has gaps separating the quasiparticle from the bands as we already expect. In the rightmost figure, the quasiparticle is absent for low chemical potentials/dopings (black, blue); increasing the doping from black to blue curve does not produce a quasiparticle but just a spectral weight transfer from low to high frequencies. This signals that the system is approaching the insulating phase (but is not insulating yet because there is no gap in the center). All local spectra are obtained by inverse-Fourier-transforming the momentum-space spectra on a 4x4 grid of points in momentum space.

The source $\mathcal{S}$ mixes different Bloch waves:

$$
\mathcal{S}_{(0)}^{(\mathbf{n})} = \left( E_{tt}^{(2)} \frac{\omega^2}{C_a^6 r^{6\gamma}} - E_{xx}^{(2)} \frac{k_x^2 + k_y^2}{C_a^2 r^{4\beta+2\gamma}} \right) \sum_{\mathbf{n}'} \psi_{(0)}^{(\mathbf{n}')}, \tag{44}
$$

and the coefficients $\hat{M}_1, \hat{M}_2$ read

$$
\hat{M}_1 = \begin{pmatrix} \frac{2\gamma\omega r^\beta + (\beta+\gamma)C_a k_x r^\gamma}{\omega r^{\beta+1} + C_a k_x r^{\gamma+1}} & 0 & 0 & -\frac{k_y}{C_a r^{\beta+\gamma}} \\ 0 & \frac{2\gamma\omega r^\beta - (\beta+\gamma)C_a k_x r^\gamma}{\omega r^{\beta+1} - C_a k_x r^{\gamma+1}} & -\frac{k_y}{C_a r^{\beta+\gamma}} & 0 \\ 0 & -\frac{k_y}{C_a r^{\beta+\gamma}} & \frac{2\gamma\omega r^\beta - (\beta+\gamma)C_a k_x r^\gamma}{\omega r^{\beta+1} - C_a k_x r^{\gamma+1}} & 0 \\ -\frac{k_y}{C_a r^{\beta+\gamma}} & 0 & 0 & \frac{2\gamma\omega r^\beta + (\beta+\gamma)C_a k_x r^\gamma}{\omega r^{\beta+1} + C_a k_x r^{\gamma+1}} \end{pmatrix}, \tag{45}
$$

$$
\hat{M}_0 = \begin{pmatrix} \frac{\omega^2}{C_a^4 r^{4\gamma}} - \frac{k_x^2}{C_a^2 r^{2\beta+2\gamma}} & 0 & \frac{\omega k_y}{C_a^3 r^{\beta+3\gamma}} + \frac{k_x k_y}{C_a^2 r^{2\beta+2\gamma}} & \frac{(\beta-\gamma)\omega k_y}{C_a \omega r^{\beta+\gamma+1} + C_a^2 k_x r^{2\gamma+1}} \\ 0 & \frac{\omega^2}{C_a^4 r^{4\gamma}} - \frac{k_x^2}{C_a^2 r^{2\beta+2\gamma}} & \frac{(\beta-\gamma)\omega k_y}{C_a \omega r^{\beta+\gamma+1} - C_a^2 k_x r^{2\gamma+1}} & -\frac{\omega k_y}{C_a^3 r^{\beta+3\gamma}} + \frac{k_x k_y}{C_a^2 r^{2\beta+2\gamma}} \\ \frac{\omega k_y}{C_a^3 r^{\beta+3\gamma}} - \frac{k_x k_y}{C_a^2 r^{2\beta+2\gamma}} & \frac{(\beta-\gamma)\omega k_y}{C_a \omega r^{\beta+\gamma+1} - C_a^2 k_x r^{2\gamma+1}} & \frac{\omega^2}{C_a^4 r^{4\gamma}} - \frac{k_x^2}{C_a^2 r^{2\beta+2\gamma}} & 0 \\ \frac{(\gamma-\beta)\omega k_y}{C_a \omega r^{\beta+\gamma+1} + C_a^2 k_x r^{2\gamma+1}} & -\frac{\omega k_y}{C_a^3 r^{\beta+3\gamma}} - \frac{k_x k_y}{C_a^2 r^{2\beta+2\gamma}} & 0 & \frac{\omega^2}{C_a^4 r^{4\gamma}} - \frac{k_x^2}{C_a^2 r^{2\beta+2\gamma}} \end{pmatrix}. \tag{46}
$$

The coefficients $E_{tt}^{(2)}$ and $E_{xx}^{(2)}$ in the source term are the expansion coefficients of the metric, given in Eqs. (A.7) in Appendix A. Fortunately, we will not need their explicit form neither in the source nor in the matrices $\hat{M}_1, \hat{M}_0$; we are just interested in the general structure of the self-energy.

From now on we employ the WKB approximation (this was also done in [27] in absence of lattice) for two reasons (1) the scaling of elements of the coefficient matrix $\hat{M}_0$ is such that they usually give a deep potential well, hence we expect to have many bound states and high quantum numbers, precisely the regime where WKB applies (2) it is difficult to make

any analytical progress without WKB because the equations are so complicated. Introducing the tortoise coordinate $s$ in the usual way [23–25] and equating the coefficient of the first derivative $\partial_s \psi$ to zero, we arrive at the condition:

$$\det\left(\hat{I}_{4\times 4} + \frac{\partial s/\partial r}{\partial_r (\partial s/\partial r)} \hat{M}_1\right) = 0\,, \tag{47}$$

which determines the tortoise coordinate as a function of $r$. In our case, keeping only the leading scaling term in $r$, we get $s(r) \sim r^{2C_a\gamma}$. This gives the Schrödinger equation in the tortoise coordinate, with no first-derivative term:

$$\left(\partial_s^2 - \left(\frac{\partial r}{\partial s}\right)^2 \hat{M}_0\right)\psi_{(1)}^{(\mathbf{n})} = \left(\frac{\partial r}{\partial s}\right)^2 \mathcal{S}_{(0)}^{(\mathbf{n})}\,. \tag{48}$$

The solution in the classically allowed region is expressed in terms of the WKB momentum $p$:

$$\det\left(p^2 \hat{I}_{4\times 4} + \left(\frac{\partial s}{\partial r}\right)^2 \hat{M}_0\right) = 0 \Rightarrow p \sim r^{-\beta}\sqrt{\omega^2 r^{2\beta} - (k_x^2 + k_y^2)r^{2\gamma}}\,, \tag{49}$$

$$\psi_{(0)\text{IN/OUT}}^{(\mathbf{n})} = \frac{1}{\sqrt{p}}e^{\pm i \int p\,dr}\,. \tag{50}$$

The two solutions above, with $\pm$ in the exponent, are just the incoming and the outgoing solution at the horizon – the two independent solutions of the IR Schrödinger equation. For the IN solution the integration limits in (50) are $\int_{r_h}^r dr$ and for OUT they are $\int_r^\infty dr$. For the next step we again follow [4] and solve the inhomogeneous equation by definition: we write down the bulk-to-bulk propagator and integrate it over the source. Expressing the bulk-to-bulk propagator from the solutions (50) and their Wronskian $W$ as

$$G_{\text{bulk}}(r,r') = \frac{1}{W}\left[\psi_{(0)\text{IN}}^{(\mathbf{n})}(r)\psi_{(0)\text{OUT}}^{(\mathbf{n})}(r')\Theta(r'-r) - \psi_{(0)\text{IN}}^{(\mathbf{n})}(r')\psi_{(0)\text{OUT}}^{(\mathbf{n})}(r)\Theta(r-r')\right], \tag{51}$$

we write $\psi_{(1)}^{(\mathbf{n})}(r) = \int dr' G_{\text{bulk}}(r,r')\mathcal{S}_{(0)}^{(\mathbf{n})}(r')$ and restrict the integration to the $\epsilon$ neighborhood of the horizon (we are deliberately sketchy here as [4] contains a detailed discussion). When everything is said and done and when we match the UV solution and read off the field theory propagator $G_R$ from (28), we obtain

$$G_R^{(\mathbf{n})} \sim \frac{1}{\omega - v_F k_\perp - i\text{Im}\Sigma^{(\mathbf{n})}}\,, \qquad k_\perp \equiv |\mathbf{k} - \mathbf{k}_F|\,, \tag{52}$$

$$\text{Im}\Sigma^{(\mathbf{n})} \sim e^{-2I^{(\mathbf{n})}} + \tilde{k}^2\left(\sum_{\mathbf{n}'} e^{-2I^{(\mathbf{n}')}}\right) + \tilde{k}^4\left(\sum_{(n_x',n_y'')} e^{-2I^{(n_x',n_y'')}}\right) + \tilde{k}^4\left(\sum_{(n_x'',n_y')} e^{-2I^{(n_x'',n_y')}}\right)$$

$$+ \tilde{k}^4\left(\sum_{\mathbf{n}''} e^{-2I^{(\mathbf{n}'')}}\right) + \dots \tag{53}$$

$$\tilde{k}^2 \equiv \frac{\delta\mu}{\mu_0}\left(E_{tt}^{(2)}\frac{\omega^2}{C_a^6 r_h^{6\gamma}} + E_{xx}^{(2)}\frac{k^2}{C_a^2 r_h^{4\beta+2\gamma}}\right)\,. \tag{54}$$

In the first line, the expansion around the Fermi surface at $\omega = 0$ and the Fermi momentum $\mathbf{k}_F$ obviously assumes that a sharp Fermi surface exists; this is true if bound states exist in the effective Schrödinger potential [46]. By $I$ we denote the integral of the WKB momentum appearing in (50) which, from (49) scales as

$$I \sim \frac{|\mathbf{k}|^{\frac{2\gamma-1}{\gamma-\beta}}}{\omega^{\frac{\beta+\gamma-1}{\gamma-\beta}}}\,, \tag{55}$$

and the notation $\mathbf{n}', \mathbf{n}'', (n'_x, n''_y)$ and the like is hopefully clear: in $\mathbf{n} \equiv (n_x, n_y)$ we have the lattice momentum $(k_x + 2\pi n_x Q, k_y + 2\pi n_y Q)$, in $\mathbf{n}'$ the momenta are shifted from $\mathbf{n}$ by $(\pm 1, \pm 1)$, in $\mathbf{n}''$ the shift is $(\pm 2, \pm 2)$, in $(n'_x, n''_y)$ the possible shifts are $(\pm 1, \pm 2)$ and so on.

The above expression for the self-energy is the main analytical result of our perturbative expansion. So how does the peak width scale with frequency? In absence of lattice, only the first term in (53) survives and from (55) we come back to the conclusion of [27] that the self-energy is exponentially small at small $\omega$ if the exponent $\beta + \gamma - 1$ is positive (i.e. if $\alpha < -1$), or else it is always of order unity, even at $\omega = 0$, and there is no sharp quasiparticle. But upon adding the zone-mixing terms, the magnitude of $\Sigma$ depends in a complex way both on $\omega$ and on the position in the $\mathbf{k}$ space. We can discern the following trends.

1. Exponential decay of $\exp(-2I^{(\mathbf{n})})$ for small $\omega$ (when $\beta + \gamma - 1$ is positive) can be partially compensated by the power-law factors in $\tilde{k}^{2n}$. True, these factors are suppressed by factors of $\delta\mu/\mu_0$ but they can add up to a non-negligible quantity. This explains the lack of a sharp peak in some cases when $\beta + \gamma > 1$, i.e. $\alpha < -1$.

2. If $2\gamma > 1$ (in particular if $2\gamma \gg 1$) and $|\mathbf{k}| < 1$ then the numerator in (55) can come closer to zero than the denominator so even for $\beta + \gamma > 1$ and very small $\omega$ the integral $I$ can still be quite small, i.e. the self-energy can be of order unity. This explains the weakening of the quasiparticle peak along the $\Gamma X M \Gamma$ trajectory (for finite $|\mathbf{k}|$) in some figures.

3. The appearance of Hubbard-like bands comes from the competition of the $\omega^{2n}$ prefactors and the exponential $\exp(-2I^{(\mathbf{n})})$ at finite $\omega$ – for small $\omega$ the influence of the prefactors is negligible[18] but for larger $\omega$ the exponential factors are all of order unity in any case, so what matters is whether $\tilde{k}^2$ grows or decays with $\omega$: from (54), the maximum (the band) is located roughly at $\omega \sim C_a^3 r_h^{3\gamma}$; of course such a maximum is broad as it comes from a sum of various power laws. Emphatically, this has nothing to do with the position of the *poles*, which the self-energy cannot influence, and the wide bumps are not poles anyway; they are finite maxima and come solely from the variation of the finite part of the propagator (large imaginary part of the self-energy means smaller spectral weight).

We have thus explained qualitatively the complex structure of the region with a quasiparticle in the "phase diagram" in Fig. 4. For now we will not attempt to understand the action of the dipole coupling $\kappa$ analytically. In leading order expansions that we use in this subsection, $\kappa$ never appears – naively one would think its influence is not important. However, $\kappa$ shifts the effective momentum and thus influences the *position of the pole*, not just the self-energy. Such effects require a more thorough analytical treatment than we are able to perform at present.

A systematic study of MDCs is also beyond the scope of this paper. Perhaps more interesting than EDCs as they show the influence of the lattice more directly, they are also far more complicated, and show strong dependence on the lattice strength $\delta\mu/\mu_0$. We have looked at a few examples in Appendix D, but we leave a proper treatment for further work.

## 5 Green functions in Matsubara frequencies: comparison to the Hubbard model?

Encouraged by the detailed phenomenological understanding that we have achieved, we explore now to what extent are our findings in line with real-world lattice models of strongly

---

[18]If $\beta + \gamma - 1 > 0$ then the exponential smallness is not much influenced by power-law prefactors; if $\beta + \gamma - 1 < 0$ then the power-law prefactors are important but they do not exist in front of the first term, $\exp(-2I^{(\mathbf{n})})$, so the self-energy remains large.

correlated metals. A natural candidate is the Hubbard model on a square lattice. This famous Hamiltonian

$$H = -t \sum_{\langle i,j \rangle} \sum_{\sigma} \left( c^{\dagger}_{i\sigma} c_{j\sigma} + c^{\dagger}_{j\sigma} c_{i\sigma} \right) + U \sum_{i} n_{i\uparrow} n_{i\downarrow}, \tag{56}$$

is a prototype model for strongly coupled electron systems, showing a wide range of experimentally relevant phenomena: strange-metallic behavior, Mottness, superconductivity, various exotic orders. The hopping is over the neighboring lattice sites, the spin is $\sigma \in \{\uparrow, \downarrow\}$, and the crucial parameter is the Coulomb interaction strength $U/t$. A review and comparison of state-of-the-art techniques can be found in [32,35]. We stick to the basic Hubbard Hamiltonian, with no additional interactions, non-nearest-neighbor hoppings etc. – since any direct relation to holography is tentative at best, it makes no sense to fine-tune the model.

The idea is to compare the holographic two-point function to the correlation functions of the quantum Monte Carlo calculation (CTINT); therefore, unlike the previous section where we looked solely at the spectral function, i.e. the imaginary part of $G_R$, here we look at the whole propagator. The object of comparison is the Green function *in Matsubara frequency* $G(i\omega_n)$. This object is the direct outcome of the CTINT method and of most quantum Monte Carlo methods (although real-time quantum Monte Carlo methods exist [66–68], they are of recent origin and not much data has been accumulated so far). A detailed study of the two-point retarded Green function in Matsubara frequencies for the Hubbard model on the square lattice has been published in [69]. For comparison with holography we have used the data from that study, provided to us by the first author of [69]. The data consists of Green function values for a range of imaginary frequencies, computed in the doped Mott insulator regime with $U/t = 10$ in a range of occupancy values $n = 0.500, 0.475, 0.450, 0.425$ (set by tuning the chemical potential), and for temperatures $T = 0.3, 0.5, 0.7$ (in units of $4t$). The occupancy $n = 0.500$ represents the half-filling. From analytical continuation, this phase is believed to have a quasiparticle or at least a clear maximum near $\omega = 0$ [36–38] at low temperatures (compared to both the chemical potential and lattice wavevector) – but for our temperature range $(0.3 - 0.7)$ one expects that the quasiparticle is already barely visible. The fit to holographic data will nevertheless show a clear quasiparticle, a puzzle that we try to resolve at the end of the section.

## 5.1 The fitting procedure

In order to express the Green functions in Matsubara frequency $i\omega_n$, we start from the real-frequency functions obtained from holography and apply the well-known Hilbert transform

$$G_R(i\omega_n) = -\frac{1}{\pi} \int_{-\infty}^{\infty} d\omega \frac{\mathrm{Im} G_R(\omega)}{i\omega_n - \omega}. \tag{57}$$

This is simply an integral over real frequencies $\omega$ and presents no difficulties, unlike the analytical continuation necessary to arrive at real-frequency results from $G(i\omega_n)$. There we see one advantage of holography, which directly produces real-frequency Green functions. Details on the numerical integration of (57) can be found in Appendix E.

Although the Matsubara Green function is mathematically well-defined for any imaginary $\omega_n$, in physical quantities at temperature $T$ only the frequencies $2\pi(n+1/2)T$ show up, so only these points were computed in CTINT and consequently we also only compute the holographic $G(i\omega_n)$ for such points. The momenta **k** are chosen along the triangular high-symmetry path ΓXMΓ for the square lattice $4 \times 4$. Although the reference [69] contains also the $8 \times 8$ data, we have found it unnecessary to use these – the deviations between the holography and the Hubbard model are larger than the systematic finite-size errors so we would get similar results even if fitting to CTINT data for a larger lattice.

In the holographic model that we employ, the free parameters are the scaling exponent $\alpha$ (we still keep $\delta$ fixed) and the conformal dimension $\Delta$, whereas in the Hubbard model the free parameter is $U/t$. We can also tune the chemical potential $\mu_0$ and the temperature $T$ but these are external parameters, which are tuned for a given Hamiltonian. The idea is that the combination $(\alpha, \Delta)$ that determines the scaling and interactions defines the effective Hamiltonian and should be fit to the Hubbard $U/t$. The inclusion of $\Delta$ may be controversial: in holography it really characterizes the probe, i.e. the dimension of the operator $O$ in the correlation function $\langle OO \rangle$ that we compute, not the dual field theory itself or its Hamiltonian. However, in a model of correlated electrons, the operator in the Green function is always the electron annihilation operator $c_{j\sigma}$ (on lattice site $j$ and with spin $\sigma$), and in holography we always consider a probe Dirac fermion, so the requirement that the latter acts as a proxy for the former is effectively a constraint on the dual field theory (that the Dirac probe really probes the degrees of freedom of an elementary electron).

How to match the temperatures and the chemical potential? To that end we need to match the *units* in the two models as $\mu_0$ and $T$ are both dimensionful quantities. This can be done in two ways. One, brute force way is to scan not only the parameters $(\alpha, \Delta)$ but also $\mu_0$ and $T$ for each $(U/t, n, T)$ triple in CTINT data. We have done this only with limited resolution as the dimension of the parameter space is quite large. The other way is to notice that the self-energy of the Hubbard model in real space (i.e. in the space of lattice sites $j, l$) $\Sigma_{jl}(i\omega_n)$ can be written as

$$\Sigma_{jl}(i\omega_n) = U\langle n\rangle \delta_{jl} + \Sigma_{\mathrm{dyn}}(i\omega_n). \tag{58}$$

The first term is the static contribution from the mean-field Coulomb repulsion and the second term is dynamical and comes from quantum corrections. This term is known to drop to zero in the static limit ($i\omega_n \to \infty$); this was the reason that we needed the dynamical cutoff $\mathcal{D}$ in holography. Therefore, only diagonal terms remain in the static limit; off-diagonal elements are zero. The structure of the real-space propagator $(G_R)_{jl}$ is now

$$[G_R(i\omega_n)]_{jl} = \left[(i\omega_n - U\langle n\rangle)I - \mathcal{H} - \Sigma_{\mathrm{dyn}}\right]^{-1}_{jl}, \tag{59}$$

where $I$ and $G_R(i\omega_n)$ are the unit matrix and the propagator in real (lattice-site) space and $\mathcal{H}$ is the nearest-neighbor hopping matrix $\mathcal{H}_{jl} = t\delta_{|j-l|,1}$. The first term in (59) contributes only to the diagonal elements of $G_R$, the last term drops to zero in the static limit, so off-diagonal we only have the contribution of the hopping matrix $\mathcal{H}$ from which we can read off $t$. The algorithm is therefore:

1. Starting from the Matsubara Green functions $G(i\omega_n, \mathbf{k})$, perform the inverse Fourier transform in momentum to get the real-space function $G_{jl}(i\omega_n)$.

2. For the largest $i\omega_n$ available in the numerics, invert the real-space matrix to get $G_{jl}^{-1}(i\omega_n \to \infty)$.

3. From (59), the terms bellow/above the main diagonal of the result equal $\pm t$.

Once we have $t$, we directly tune $T$ and $\mu$ for computing $G_{\mathrm{AdS}}$ to the same values as in CTINT (in the latter they are in units of $4t$). Since the holographic model is certainly not exactly the Hubbard model (at best it comes close), and may well have long-range interactions, the above procedure is not exact, but it serves as a decent estimate, and yields similar results as the brute force fit.

One last technical remark is in order before we describe the fitting procedure. In AdS/CFT literature, the spectral function is usually defined as the imaginary part of the trace of the retarded Green function: $A(\omega, \mathbf{k}) = -(1/\pi)\mathrm{Tr}\,\mathrm{Im}\,G_R(\omega, \mathbf{k})$.[19] This quantity is basis-invariant;

---

[19]Depending on convention the prefactor $-1/\pi$ might be different, but that is not essential now.

it does not depend on the basis chosen to describe the spinor. On the other hand, Hubbard model correlation functions obtained by quantum Monte Carlo are typically expressed in the spin basis, so in absence of magnetic field they are of the form $G_R = \mathrm{diag}(G_{11}, G_{22})$ with $G_{11} = G_{22}$. For that reason, the notation $G_R$ for the Hubbard model usually means $(G_R)_{\sigma\sigma}$ with $\sigma$ up or down all the same. We could write everywhere in this section $\mathrm{Tr}G_{\mathrm{AdS}}$ on one hand and $(G_{\mathrm{CTINT}})_{\sigma\sigma}$ on the other but that would clutter the notation too much. We have decided to use solely the trace, so both in real and imaginary frequencies we write the trace in front of $G$ both for AdS/CFT and CTINT functions. Although the trace is not explicitly performed in CTINT, we know that this is the invariant quantity behind the element $G_{11} = G_{22}$ of the CTINT calculation. The only trouble is the mismatch in normalization: in CTINT conventions, the AdS/CFT result should be normed not to unity but to 2. We have corrected for this by explicitly renormalizing the AdS result by $1/2$. As a result, we compare the same type of Green functions.

Now we describe the fit of the intrinsic parameters of the model $(\alpha, \Delta)$. We look for the combination which minimizes the merit function. The merit function $M_d$ is just the normalized deviation of weight $d$ between the holographic function $G_{\mathrm{AdS}}$ and the CTINT function $G_{\mathrm{CTINT}}$:

$$M_d \equiv \frac{1}{N^d} \sum_{n,j} |\mathrm{Tr}\delta G\left(i\omega_n, \mathbf{k}_j\right)|^d = \min.$$

$$\delta G \equiv G_{\mathrm{AdS}}\left(i\omega_n, \mathbf{k}_j; \alpha, \Delta\right) - G_{\mathrm{CTINT}}\left(i\omega_n, \mathbf{k}_j; U/t\right), \quad N \equiv \sum_{n,j} 1. \tag{60}$$

We have tried different weights $d = -2, -1, 1, 2$, with very similar results.[20] The solution corresponds to the minimum of $M_d$. As we discussed above, by $G_{\mathrm{AdS}}$ and $G_{\mathrm{CTINT}}$ we mean the full $2 \times 2$ matrices. The trace in front is the trace in the internal space of these matrices.

If we turn on also the dipole coupling $\kappa$, we should in principle perform a three-dimensional scan. In order to avoid that, we perform the scan in $(\alpha, \Delta)$ for a few $\kappa$ values between 0 and 5 (for larger $\kappa$ the gap is too broad and clearly far from the Hubbard data), find that the $\kappa$ value does not influence much the position (but only the depth) of the minimum in the $(\alpha, \Delta)$ plane, and then do a detailed scan in $\kappa$ just near these minima.

In hindsight, quantitative accuracy of the fit is not a great concern as the main limiting factor is the transition from $G(\omega)$ to $G(i\omega_n)$; the integral (57) erases a lot of information so several rather different solutions may all give a very good fit in Matsubara domain. Until we can find a different object for comparison (perhaps real-time $G_R$ from real-time quantum Monte Carlo), it does not make sense to go for a high-resolution scan of the parameter space.

## 5.2 The solution

We first fix $\kappa = 0$ and perform the fit in the $(\alpha, \Delta)$ space only. The motive is that it is useful to see if $\kappa$ is essential or not for reproducing the Hubbard model results, and to know how much can we get from the minimal model. In Fig. 12 we plot the merit function $M_2$ for a range of $(\alpha, \Delta)$ values. The landscape contains a few local minima, one of which is rather dominant as the global minimum at

$$(\alpha, \Delta)_{\mathrm{sol}} = (-1.3 \pm 0.1, 1.20 \pm 0.05).$$

While in this case the global minimum is quite dominant, it is still in fact only by a factor of 5 better than the next smallest local minimum.

With nonzero dipolar coupling, the outcome is the Fig. 13. The local minima are pretty much in the same places as in Fig. 12 but the depths differ somewhat. In particular, the global

---

[20]It is often said that negative $d$ works better, imposing strict constraints on large-$i\omega_n$ properties, which translate to the normalization of the real-$\omega$ spectral function, but we have seen little difference in our fits no matter what $d$ is.

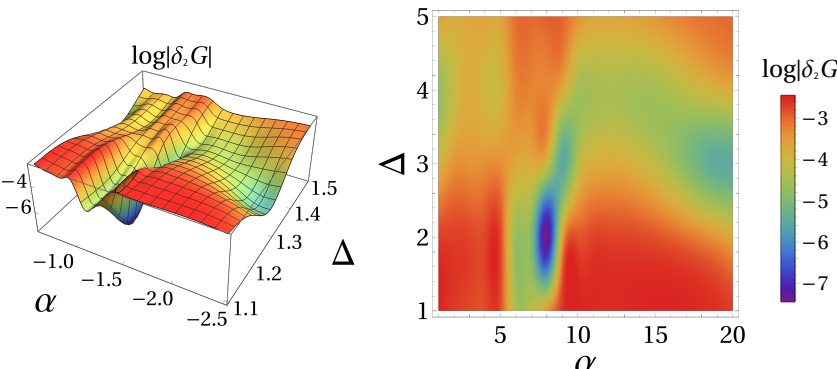

Figure 12: The squared norm of the difference $|\mathrm{Tr}\delta G|^2$ ($M_2$ function from (60)) between the holographic and CTINT Matsubara Green functions. For better viewing, we give both a 3D plot (left) and a density plot (right) with the same data. Several local minima are visible, but there is also a rather clear global minimum around $(\alpha, \Delta) = (-1.3, 1.2)$.

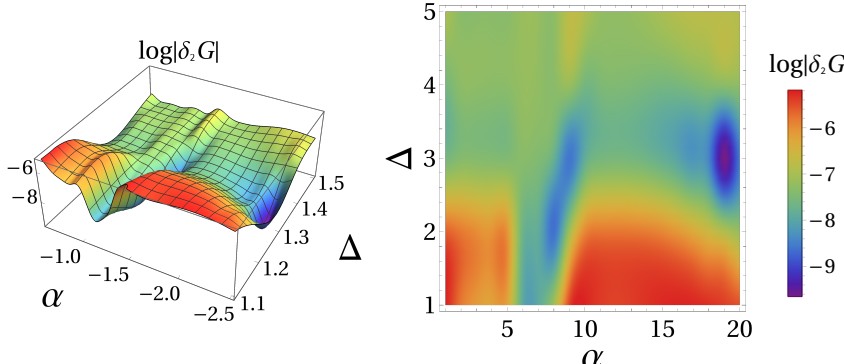

Figure 13: Same as Fig. 12 (squared norm of the difference $|\mathrm{Tr}\delta G|^2$ between the holographic and CTINT Matsubara Green functions) but for nonzero $\kappa = 1.5$. We have picked the $\kappa$ value which yields the global minimum of the deviation (see the text for details of the fitting procedure for nonzero $\kappa$). The local minima remain in the same positions as for $\kappa = 0$ but their depths and consequently the position of the deepest minimum change; the global minimum is now at $(\alpha, \Delta) = (-2.3, 1.5)$.

minimum is now different and lies at

$$(\alpha, \Delta, \kappa)_{sol} = (-2.3 \pm 0.1, 1.30 \pm 0.05, 1.5 \pm 0.5).$$

The relatively large uncertainty on $\kappa$ is simply due to the fact that once we move away from $\kappa = 0$ (and until we reach large values, around $4-5$), the solution is not very sensitive to $\kappa$. Therefore, even though the overall structure of the landscape is similar with $\kappa = 0$ and $\kappa > 0$, the minima can go up and down when $\kappa$ becomes nonzero. Therefore, it is important to decide if $\kappa$ should be fixed to zero (presumably by some symmetry) or not. Notice that both solutions are in the violet zone of the "phase diagram" in Fig. 4, with a quasiparticle (or at least something similarly-looking), separated from the upper and lower band by a soft gap.

The comparison of $\mathrm{Tr}G(i\omega_n)$ for AdS/CFT and Hubbard model is given in Fig. 14. In addition to the real and imaginary part of the propagator, we give also the absolute value of their difference. Notice that in the large-frequency regime the curves almost fall on top of each other as they should because the asymptotics is universal. At smaller frequencies, the real

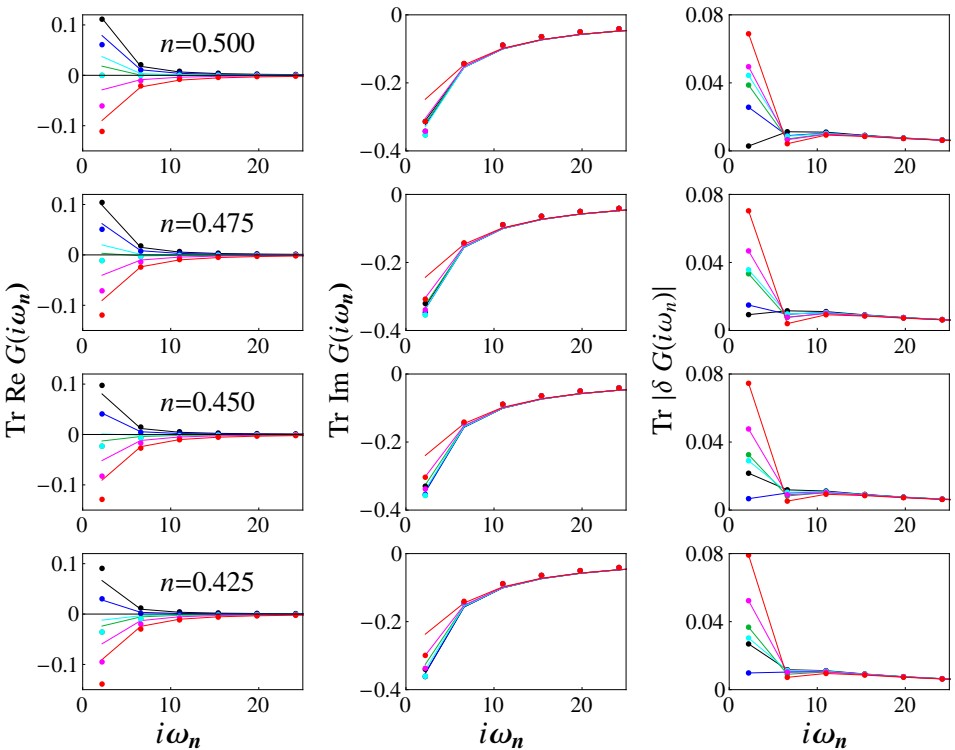

Figure 14: Real part (left) and imaginary part (middle) of the Matsubara Green function $\mathrm{Tr} G(i\omega_n)$, for the holographic fermion (solid lines) and for the QMC solution of the Hubbard model (circles), together with the module of the deviation between the two $|\mathrm{Tr}\delta G(i\omega_n)|$ (right; the lines are just to guide the eye). The functions are computed for the canonical Matsubara frequencies $i\omega_n$ and for six standard momentum values $((0,0)$ black, $(\pi/2,0)$ blue, $(\pi,0)$ cyan, $(\pi,\pi/2)$ magenta, $(\pi,\pi)$ red, $(\pi/2,\pi/2)$ green) along the high-symmetry line ΓXMΓ of the unit cell. The holographic model is for the parameters yielding the best fit to the QMC data: $(\alpha,\Delta) = (-1.3\pm0.1, 1.20\pm0.05)$, with no dipole coupling $(\kappa = 0)$. The temperature is $T = 0.7$.

part generally fares better than the imaginary part. Overall, there is a good agreement. The agreement is even better when $\kappa$ is allowed to vary (though of course increasing the number of fit parameters nearly always yields better fits), as seen in Fig. 15. At the $(\pi,\pi)$ point the agreement is much worse, just like for $\kappa = 0$. At first glance, such an effect could be discarded as a mere artifact, but in the near-horizon analysis in subsection 4.2, we have shown below Eqs. (54-55) that special behavior near the corners of the Brillouin zone is in fact expected. However, this phenomenon is absent in the Hubbard model, hence this curve is never very well fit to the CTINT data.

While we need the Matsubara propagators for comparison, it is the real-frequency propagator which is of prime physical importance. The solution without dipole coupling is shown in Fig. 16, with momentum-integrated spectra $A_{\mathrm{loc}}(\omega)$ (top) and momentum-resolved spectra $A(\omega,\mathbf{k})$ (bottom). The overall structure of the spectrum in this regime is in fact known from previous subsections – quasiparticle plus the bands. The interesting fact, best seen from the integrated spectra, is that the chemical potentials corresponding to the four doping values of the Hubbard Hamiltonian are very close – the peak is barely moving upon dialing $n$. The chemical potentials (from highest to lowest $n$) are $\mu_0 = (1.21, 1.22, 1.24, 1.25)\pm 0.01$, i.e. quite close to each other. A good fit as in Fig. 14 is obtained thanks to the large weight of the peak – slow

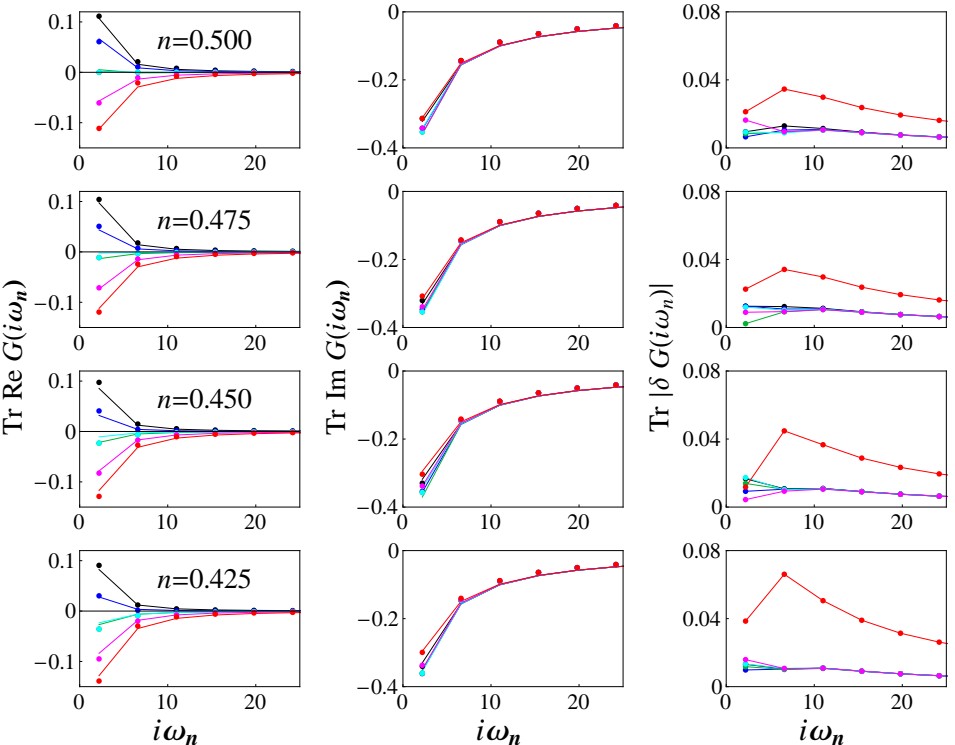

Figure 15: Same quantities as in Fig. 14 (real and imaginary part of $\text{Tr}G(i\omega_n)$ for AdS/CFT and Hubbard models, and the module of the difference between the models) but with nonzero dipole coupling $\kappa$, for the three parameters yielding the best fit to the QMC data: $(\alpha, \Delta, \kappa) = (-2.3 \pm 0.1, 1.20 \pm 0.05, 1.5 \pm 0.5)$. The inclusion of dipolar coupling makes the gap between the bands and quasiparticle more prominent and allows for a better fit.

movement across the $\omega$-axis is compensated by the fact that a lot of weight is moving, hence the broad interval of $\text{Re}G(i\omega_n)$ values at small $i\omega_n$ (which roughly corresponds to the spectral weight asymmetry in real frequencies) is still reproduced. We will come back to the question how physical this solution is (in other words, is this really what is happening in the Hubbard model in order to produce the resultant Matsubara curves). The fits for the remaining two values of the temperature are given in Appendix E.

Finally, in Fig. 17 we appreciate the overall shape of the dispersion curve in the density plot for an intermediate temperature (A) and for a very high temperature $T = 10$ at six standard momenta (B). The lower temperature is not much of a surprise as we have already seen the general properties of $A(\omega, \mathbf{k})$ in Fig. 10. But the high-temperature plot provides another hint toward Hubbard physics: the curves evolve toward a Gaussian in $\omega$, which is believed to happen also in the Hubbard model in the high-$T$ limit [70]. We do not give the full density plot here as we have already demonstrated the temperature evolution in density plots in Fig. 7, and in particular because we want to compare to a Gaussian, which is easier to visualize for just a few momentum slices.

What is the physical significance of the solutions found? The fits are quite good and the landscape of the merit function (Figs. 12 and 13) is robust in the sense that the same 2-3 local minima always appear. However, these minima can become higher or lower when turning on $\kappa$. The spectra corresponding to these minima all contain the quasiparticle and the bands but in detail they differ. What is most suspicious when looking at Figs. 12 and 13 is that on average *nearly all points in the parameter space* have relatively small deviations from the CTINT

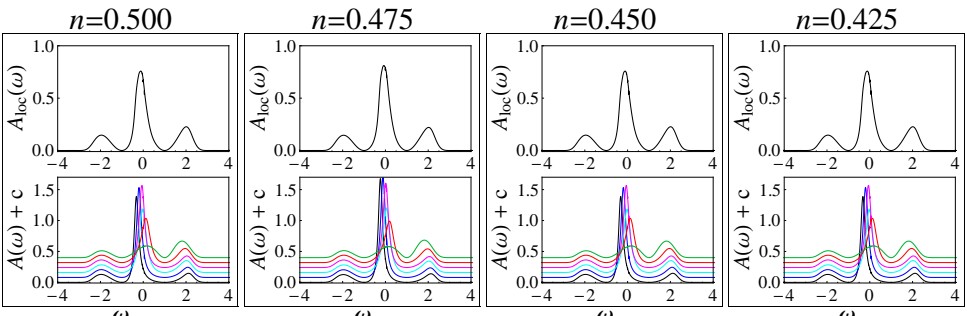

Figure 16: The local or momentum-integrated (top panel) and the momentum-resolved (bottom panel) spectral weight for the best fit of the holographic spectral function to the quantum Monte Carlo data for the Hubbard model, yielding the parameters $(\alpha, \Delta, \kappa) = (-2.3, 1.3, 1.5)$. The local spectrum shows the by now familiar structure lower band + quasiparticle + upper band. What is unexpected is that the chemical potentials for the four doping values are very close for each other: $\mu_0 = 2.10, 2.25, 2.40, 2.50$ (left to right), and indeed the local spectral function barely changes between $n = 0.500$ and $n = 0.425$. Similar conclusions follow also from the momentum-resolved spectra in the bottom row. The rough explanation is that the strong quasiparticle in the center can transfer a lot of spectral weight from $\omega < 0$ to $\omega > 0$ even for a minimal change of chemical potential.

solution and present relatively good fits! This suggests that the merit function is not very well chosen. Physically, the behavior we see in the spectra is expected for these values of $n$ and $U/t$ but only at much lower temperatures [37, 39]. We do not understand the origin of this systematic mismatch – when we determine the temperature in AdS/CFT by brute-force fitting we may well have large errors but we have also checked the units of temperature directly, by estimating the effective $t$ value in holography. It seems that the temperature mismatch is physical and that the holographic system reacts differently to a heat bath. Another possibility is that the real-time spectra obtained by analytic continuation (e.g. in [37, 39]) contain large errors but that is less likely as many authors obtain similar results. We hope to understand this better in the future.

## 6 Discussion and conclusions

In our photoemission adventure we have gained a lot of knowledge on the phenomenology of holographic spectra on 2D lattices; we have also developed a few useful tools on the way. On the other hand, there is no sharp, golden-bullet conclusion on how exactly all of this relates to real-world systems, apart from the basic features – non-Fermi liquids, asymmetric and momentum-dependent self-energies, Hubbard-like bands and some (limited) signs of Mottness. More specifically, we have found the following robust features:

1. The quasiparticle peak, the bands and the strong momentum dependence of the spectrum are the three possible robust features of photoemission spectra on the hyperscaling-violating ionic lattice; they are not all present for all parameters, but may or may not be there as shown in the phase diagrams in Fig. 4. In absence of a quasiparticle, the central peak remains but becomes a wide bump, resembling a strange metal regime.

2. The quasiparticle pole is of similar origin as usual in holographic systems, stemming from a zero in the holographic source at the boundary. But the bands are novel – they

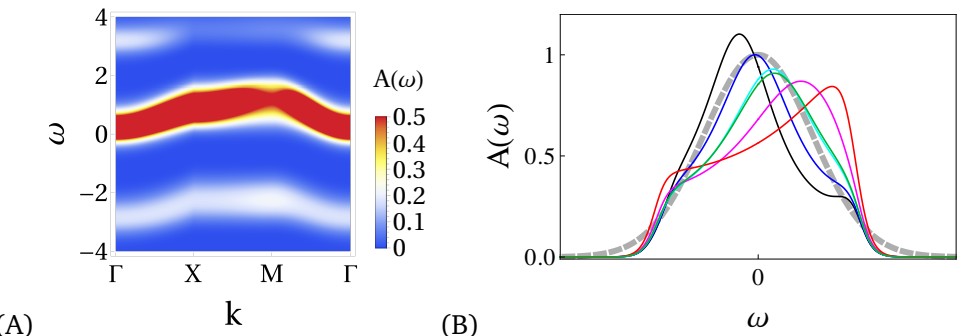

Figure 17: (A) Density plot for the spectral function $A(\omega, \mathbf{k})$ for the standard momentum set ($(0,0)$ black, $(\pi/2, 0)$ blue, $(\pi, 0)$ cyan, $(\pi, \pi/2)$ magenta, $(\pi, \pi)$ red, $(\pi/2, \pi/2)$ green), for the best Hubbard fit with dipole coupling: $(\alpha, \Delta, \kappa) = (-2.3 \pm 0.1, 1.3 \pm 0.1, 1.5 \pm 0.5)$, at temperature $T = 0.5$. The central quasiparticle peak together with the upper and lower band is clearly seen, and the gaps between the bands and the center are rather hard. (B) Spectra $A(\omega)$ for the standard set of momenta (solid curves, black to green) for the same background but with $T = 10$. At very high temperatures the spectrum evolves toward a single broad peak with a Gaussian profile – the gray dashed line is a Gaussian centered at the chemical potential $\omega = 0$. The fit is best for the blue curve, which has its maximum approximately at $\omega = 0$; for the others, we do not expect a zero-centered Gaussian to be a good fit except at large $\omega$, which is precisely what we see.

can exist (separated from the central peak by soft gaps) even in absence of dipole coupling and arise from the umklapp terms in self-energy, which make its $\omega$ dependence nonmonotonic.[21]

3. Inhomogeneous and anisotropic (i.e., momentum-dependent) quasiparticle weight is also explained through the near-horizon analysis of umklapp terms in the self-energy. While it is somewhat at odds with the Hubbard model, it might have a role in some realistic strange metals.

4. The influence of the dipole coupling is to make the side bands more robust and the corresponding gaps harder. At larger couplings a soft central gap also appears, giving rise to a Mott-like regime (but the gap never fully opens for all momenta). We do not understand this analytically but it is likely related to the effective momentum shift and the zero-pole duality found for $\kappa$ in homogeneous systems in [45, 59, 61].

5. The fit to the Hubbard model is surprisingly good, given the vast differences in microscopic physics from the holographic model. While the anisotropy of the spectrum always spoils the fit, the general structure is preserved, and the dipole coupling $\kappa$ further helps to come closer to the Hubbard model. However, the fact the we have multiple local minima of the merit function, often with rather different real-frequency shapes, suggests that good fits to *Matsubara* functions do not mean that much.

In relation to the last point, we conclude that, quite generally, explicit fitting of the holographic Matsubara Green function to that of the quantum Monte Carlo calculations is not the best way to match the two. Essentially, we have attempted to circumvent the well-known unreliability of analytic continuation from imaginary to real frequencies by calculating the real-frequency

---

[21]We thank Jan Zaanen for asking this question, in particular how much the existence of bands hinges on the dipole coupling; it turns out that the dipole coupling strengthens the bands but is not a necessary condition.

functions directly (quite a natural thing to do in holography); in this way we work with perfectly reliable real-frequency functions, but when we try to compare to the Matsubara data from CTINT calculations, the problem shows up, just in reverse – it is easy to get a good fit but we do not know how relevant that is when it comes to real-frequency propagators. Still, the AdS/CFT curves pass several additional tests for the Hubbard physics (e.g. the single Gaussian limit at high temperatures), so we believe it makes sense to connect holographic models to lattice model Hamiltonians, but one has to find better merit functions and more appropriate objects for comparison. We would be interested to try with real-frequency Green functions obtained from the recently developed real-time Monte Carlo methods [66–68]. And it certainly makes sense to construct AdS duals of simpler and lower-dimensional models, i.e. an Anderson impurity, in a controlled way. Although it is unreasonable to expect that typical condensed matter lattice models have a controlled, microscopic gravity dual at all, work in this direction is still useful, as it can tell us which properties of these models are important.

On the technical side, quite a number of goodies employed in this work are likely useful also in different contexts. In particular, we have learned how the explicit ionic lattice influences the spectrum at finite Lifshitz $\zeta$, i.e. for a general case of the "holographic scaling atlas" of [29,30]. The IR analysis with the matching to the UV that we have performed is in principle feasible in any background; it would be exciting to apply it to other AdS lattice models such as scalar lattices or the spontaneously generated charge density wave lattices (stripes, checkerboards and the like). These lattices are relevant also in the IR and we may expect further surprises. The UV cutoff that we employ, following [43] but making the cutoff fully holographic, is also a viable method for many purposes, allowing us to model multi-scale systems by coupling many AdS spaces in a similar vein as in [64].

On the qualitative, phenomenological side, our findings seem robust and in line with the general knowledge on strange metals [33,49,52]. The million dollar question is – is holography ready to answer specific condensed matter questions, i.e. to teach us something about a specific class of systems (like strange metals) rather than just general phenomenology as in this paper. It seems we are not there yet although we are much closer than a decade ago – papers like [52] as well as recent applications to transport phenomena have brought AdS/CFT closer to the lab than ever before. Our present work does not answer any smoking gun question in condensed matter, but it teaches us where to look further (other types of lattices, stronger lattices, near-horizon analysis like in this paper, various hyperscaling-violating systems as very robust and general) when trying to formulate and solve deep questions in quantum matter within the holographic framework.

## Acknowledgments

We thank Jakša Vučičević for endless discussions and a wealth of ideas and information on strange metals and the Hubbard model; we also thank Jakša for providing us with the CTINT results for the Hubbard model. We are grateful to Jan Zaanen, Koenraad Schalm, Nicolas Chagnet, Aristomenis Donos and Mark Golden for helpful discussions and comments. We thank two anonymous referees for constructive and stimulating comments. This work has made use of the excellent Sci-Hub service. Work at the Institute of Physics is funded by the Ministry of Education, Science and Technological Development and by the Science Fund of the Republic of Serbia, under the Key2SM project (PROMIS program, Grant No. 6066160).

## A  Coefficients of the IR expansion

At zeroth order (homogeneous solution), the scaling exponents (reprinted here for convenience from (10)) and the coefficients read (with $C_{rr}^{(0)} = 1/C_{tt}^{(0)}$ and $C_{xx}^{(0)} = 1$):

$$\beta = \frac{(\alpha + \delta)^2}{4 + (\alpha + \delta)^2}, \quad \gamma = 1 - \frac{2\delta(\alpha + \delta)}{4 + (\alpha + \delta)^2}, \tag{A.1}$$

$$\eta = -\frac{2(\alpha + \delta)}{4 + (\alpha + \delta)^2}, \quad \xi = \alpha\sqrt{\beta(1 - \beta)} + \beta + \frac{5}{4}, \tag{A.2}$$

$$C_{tt}^{(0)} = \frac{V_0\left(1 + (\alpha + \delta)^2\right)^2}{2\left(1 + 2\alpha(\alpha + \delta)\right)\left(1 + (3\alpha - \delta)(\alpha + \delta)\right)}, \quad C_{xy}^{(0)} = C_{xr}^{(0)} = 0. \tag{A.3}$$

Now we can also explicitly relate the temperature to $r_h$ and the parameters of the model:

$$T = \frac{V_0}{4\pi} \frac{\left(1 + \alpha^2 - \delta^2\right)\left(1 + (\alpha + \delta)^2\right)}{\left(1 + 2\alpha(\alpha + \delta)\right)\left(1 + (3\alpha - \delta)(\alpha + \delta)\right)} r_h^{2\gamma - 1}. \tag{A.4}$$

At first order, all inhomogeneous corrections are zero, but we can determine the exponents $\lambda$ and $\nu$ of the off-diagonal metric terms and the homogeneous corrections:

$$\lambda = \frac{(\alpha + \delta)^2}{1 + (\alpha + \delta)^2}, \quad \nu = \frac{1 + 3(\alpha + \delta)^2}{2 + 2(\alpha + \delta)^2}, \quad C_{xx}^{(1)} = C_{xr}^{(1)} = \frac{1}{1 + (\alpha + \delta)^2}, \tag{A.5}$$

whereas $C_{xy}^{(0)}$ remains undetermined and the other first-order coefficients are all equal zero. Only at second order there are $(x, y)$-dependent corrections:

$$E_{tt}^{(2)} = F_{tt}^{(2)} = \frac{\pi}{1 + (\alpha + \delta)^2}, \quad E_{xx}^{(2)} = F_{xx}^{(2)} = -\frac{2\pi(1 + 2\alpha^2 + 2\alpha\delta)}{(1 + (\alpha + \delta)^2)^2}, \tag{A.6}$$

$$E_a^{(2)} = F_a^{(2)} = -\frac{\pi^2(\alpha + \delta)}{2\alpha\left(1 + (3\alpha - \delta)(\alpha + \delta)\right)\left(-1 + 2\delta(\alpha + \delta)\right)}. \tag{A.7}$$

This tells us two important things: (1) at leading order there are indeed no inhomogeneous terms and the scaling solution remains valid, which is also consistent with the numerical integrations and theoretical expectations from the literature (2) the fact that an inhomogeneous branch exists at higher order means that it is possible to match the IR solution to the UV asymptotics discussed in Eq. (12). We have not tried to do this matching explicitly but again the fact that numerics yields a solution de facto justifies this logic.

## B  Summary of the numerics

Throughout most of the paper, we solve both the Einstein-Maxwell-dilaton system and the probe Dirac equation with a pseudospectral collocation grid solver, based on the algorithms in [50] and [51]. The pseudospectral method, by now pretty standard in works on holographic lattices, is based on a series expansion of the unknown functions $Y(x)$ in some basis,[22] and a set of points $x_i$ associated with that basis, so that the (truncated) expansion of the unknown functions exactly satisfies the differential equation at the points from this set. This idea is explained very well in [51], and the usual basis choices (that we also adopt) are the Chebyshev

---

[22]We denote schematically the independent variable(s) by $x$ and the unknown function(s) by $Y(x)$. In our case $x$ is a triple of coordinates and $Y$ is a whole set of functions but for the sake of brevity we write $Y$ and $x$ as scalars in this brief summary.

polynomials for nonperiodic directions and the Fourier series for the periodic directions. The collocation algorithm is a specific realization of the pseudospectral idea where one stores the information on the series expansion of $Y$ not as a set of series coefficients but as an array of values $Y_j = Y(x'_j)$ at some set of points $x'_j$ (the sets $x_i$ and $x'_j$ are distinct). In this way one can show (e.g. [50] and references therein) that, for a set of so-called cardinal functions $C_j(x)$, the unknown function is approximated as

$$Y(x) \approx \sum_j Y(x'_j) C_j(x), \tag{B.1}$$

thus a linear system $\hat{L} Y = f$ where $\hat{L}$ is some differential operator and $f$ are the sources, discretizes as

$$L_{ij} Y_j = f_i, \quad L_{ij} = \hat{L} C_j(x)|_{x=x'_i}, \quad Y_j = Y(x'_j), \quad f_i = f(x'_i). \tag{B.2}$$

The advantage is that the differentiation of the cardinal functions is very cheap: their values at the collocation points $x'_i$ are tabulated for pretty much any meaningful basis and can be found in the appendices of [50]. The solution of a *linear* system then reduces to the inversion of $L$. The boundary conditions (Dirichlet or Neumann) along the radial ($z$) direction are enforced by replacing the first and last row in $L_{ij}$ with the appropriate row vectors implementing the boundary condition (just as explained in [51]). Along $x, y$, the boundary conditions are periodic, so when using periodic basis functions they are automatically satisfied and no explicit boundary conditions are needed.

We use the Gauss-Lobatto grid, which consists of the extrema of the basis polynomials and the endpoints of the interval. The alternative is the Gauss-Chebyshev grid, consisting of zeros of the basis polynomials and no endpoints. For equations of motion in AdS, we consistently find the Gauss-Lobatto grid to be superior, as Gauss-Chebyshev is much more prone to artificial oscillations. This is apparently a special property of AdS: the general convergence properties of the two grids are known to be the same [50].

Since the EMD system is nonlinear, one further necessary ingredient is the Newton-Raphson iterative algorithm, i.e. expanding the nonlinear system $\hat{F}(Y) = f$ to first order in the variations $\delta Y^{(0)}$ around some initial guess $Y^{(0)}$ as

$$J^{(0)} \delta Y^{(0)} = -\hat{F}\left(Y^{(0)}\right) + f, \quad J^{(0)} \equiv \frac{\partial \hat{F}}{\partial Y}|_{Y=Y^{(0)}}. \tag{B.3}$$

This is a linear equation in $\delta Y^{(0)}$ that can be solved by a linear collocation solver by inverting the discretized Jacobian $J^{(0)}_{ij}$ in (B.3). We then update the function as $Y^{(0)} \mapsto Y^{(1)} \equiv Y^{(0)} + \delta Y^{(0)}$ and repeat the procedure until subsequent approximations are close enough in some norm: $\|Y^{(n)} - Y^{(n-1)}\| < \epsilon$. The matrix inversion is the costliest part of the whole procedure and we found it more efficient to adopt the Broyden's method instead [71, 72]. The idea here is to compute the inverse of the Jacobian *in the first iteration only* and in subsequent iterations to update directly the inverse, rather than the Jacobian itself. Denoting the inverse of the Jacobian in the $n$-th iteration by $I^{(n)} \equiv \left(J^{(n)}\right)^{-1}$, the secant formula and the matrix inversion lemma (Sherman–Morrison–Woodbury lemma) yield the equation

$$I^{(n+1)} = I^{(n)} + \frac{1}{\text{Tr}\left[\left(\delta Y^{(n)}\right)^T \cdot I^{(n)} \cdot \delta F^{(n)}\right]} \left(\delta Y^{(n)}\right)^T \cdot I^{(n)} \cdot \left(\delta Y^{(n)} - I^{(n)} \cdot \delta F^{(n)}\right), \tag{B.4}$$

with

$$\delta F^{(n)} = F\left(Y^{(n+1)}\right) - F\left(Y^{(n)}\right). \tag{B.5}$$

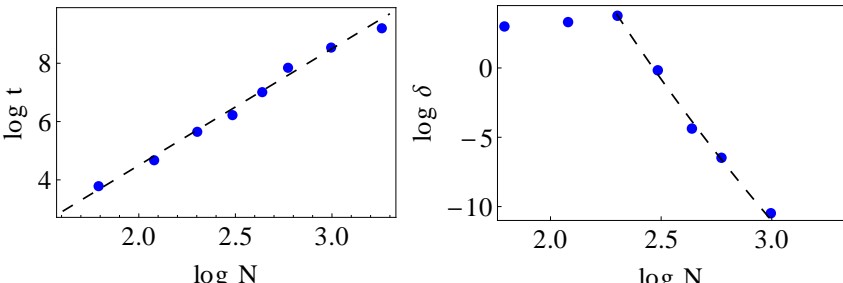

Figure 18: The running time in seconds (left) and the relative accuracy, i.e. relative deviation from the result on the largest lattice, here with $N = 26$ (right), for a number of lattice sizes. The running time grows as a power law, and the error drops sharply once we are over some critical lattice size (here about $N = 12$) and the finite-size effects stop.

Therefore, once we have $I^{(0)}$, everything proceeds by matrix multiplications only, with no further inversions. In this way the first iteration takes most time, and subsequent ones are much faster. An alternative formula is

$$I^{(n+1)} = I^{(n)} + \frac{1}{\|\delta F^{(n)}\|^2} \left(\delta F^{(n)}\right)^T \cdot \left(\delta Y^{(n)} - I^{(n)} \cdot \delta F^{(n)}\right), \tag{B.6}$$

which is generally believed [71] to be inferior to (B.4), but in some (infrequent) cases we find it actually more stable. Another possible method is to replace the inverse of the Jacobian by pseudoinverse, found by singular value decomposition (which can be programmed manually or invoked as the Mathematica function PseudoInverse). The decomposition runtime with pseudoinverse scales as $mN^4$ for an $N$-point grid with $m$ iterations, unlike the naive inverse which scales as $mN^6$ and the Broyden's method which scales as $N^6 + (m-1)N^2$. Pseudoinverse is thus the fastest method but it is often not accurate enough, eventually producing large errors, so we avoid using it.

We demonstrate the convergence properties of the solver in Fig. 18 by plotting the running time and the relative error (with respect to the highest-resolution result) for a number of resolutions, for a test EMD equation.

In practice, we start by first fixing the gauge through the Einstein-DeTurck trick, explained in [51, 73] and references therein. It makes the Einstein equations elliptic which is necessary to guarantee the numerical convergence. Then we usually perform a low-resolution run for typical parameter values on a desktop computer. Knowing the qualitative properties of the solution, we run a higher resolution calculation on a cluster for a range of parameter values of the EMD system of equations. Finally, we use the outcome as the background for solving the Dirac equations. The Dirac equations are solved in a single step of the pseudospectral solver, without iterations, as they are linear.

The semiholographic calculations (which are discussed in the next Appendix and which were not used in the main part of the paper but have proved valuable for a quick initial screening of the parameter space) were all performed on a desktop computer. All codes are implemented with double precision arithmetic.

# C Holography vs. semiholographies

While the fully consistent ionic lattice obtained by numerical solution of the partial EMD system is certainly preferable, we have experimented with two simpler approximations. We dub

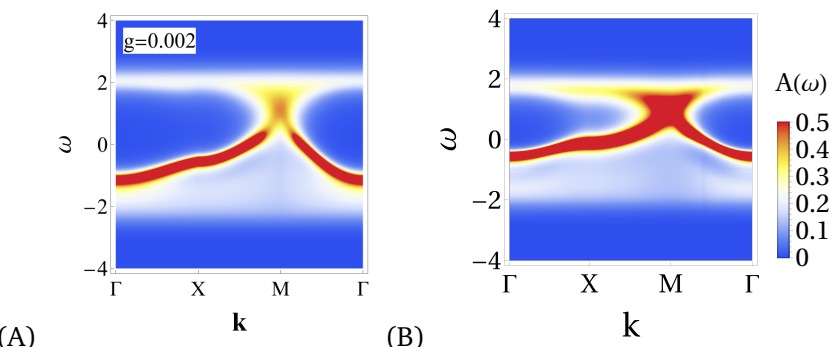

Figure 19: Semiholographic RPA Green function (A) and fully holographic Green function (B) for holographic parameters $(\alpha, \Delta) = (-1.40, 1.3)$. The hybridization for the semiholographic function is tuned to $g = 0.002$ in order to obtain a good approximation to the consistent holographic lattice calculation. As we see, for the appropriate hybridization values, semiholographic spectra reproduce all qualitative and even some quantitative features of the fully holographic spectra.

both semiholography, according to the paper [19] which pioneered this approach, however in detail they are quite different. The first is the true semiholography of the aforementioned paper, recently applied in [52] to experimental ARPES curves, and also in a hybridization-based theoretical model of strange metals [74, 75]. To remind, the idea is to introduce a linear coupling between the holographic propagator $\mathcal{G}_R$ in a homogeneous background and a free electron on the lattice. The total action is then

$$S = S_{\text{AdS/CFT}}\left(\Psi^\dagger, \Psi\right) + \int dt \int d^2 x \, \chi_{\mathbf{k}}^\dagger (i\partial_t - \epsilon(\mathbf{k})) \, \chi_{\mathbf{k}} + g \sum_{\mathbf{K}} \left(\chi_{\mathbf{k}}^\dagger \Psi_{\mathbf{k}+\mathbf{K}} + \Psi_{\mathbf{k}+\mathbf{K}}^\dagger \chi_{\mathbf{k}}\right). \quad (C.1)$$

Here, $S_{\text{AdS/CFT}}$ is the action for the holographic fermion $\Psi$ (in absence of lattice), $\chi$ is the free fermion on a square lattice with dispersion $\omega = \epsilon(\mathbf{k})$ and $g$ is the hybridization between the two.[23] The resulting dressed retarded propagator of the lattice fermion $\mathcal{G}_R$ reads (in the RPA approximation):

$$\mathcal{G}_R(\omega, \mathbf{k}) = \frac{1}{\mathcal{G}_{0R}^{-1}(\omega, \mathbf{k}) - g^2 G_R^{-1}(\omega, \mathbf{k})} = \frac{1}{\omega - \epsilon(\mathbf{k}) - g^2 G_R^{-1}(\omega, \mathbf{k})}, \quad (C.2)$$

where $\mathcal{G}_{0R}$ is the bare propagator of the same fermion $\chi$ (containing a quasiparticle pole), and $G_R^{-1}$ is the inverse holographic propagator which now has the role of self-energy. We assume that $g$ is real, i.e. $g = g^*$. Varying $g$ and the holographic parameters $(\alpha, \Delta)$ yields results similar to the full lattice calculation. We have *not* used this method neither when exploring the AdS model in Section 4 nor when fitting to the Hubbard model in Section 5. We comment upon it here because the resulting spectra look quite reasonable (we give one example in Fig. 19), and they provide a decent (though uncontrolled) approximation to the full lattice spectra.

Interestingly, the hybridization $g$ also acts as a proxy of doping in the Mott-insulator-like regime, controlling the spectral weight transfer in absence of a quasiparticle. As an example we show a few spectra in Fig. 20 which all contain a soft gap around $\omega = 0$ with two bands around $\omega = \pm 4$. As we pump up the coupling $g$, the spectral weight passes from the lower to the upper band, while the gap remains approximately open. The gap-forming mechanism is different than in fully holographic lattices: the pole in the free propagator $\mathcal{G}_{0R}$ and the pole

---

[23]Of course, one should not confuse the coupling $g$ with the determinant of the metric; it will always be clear from context which one we mean.

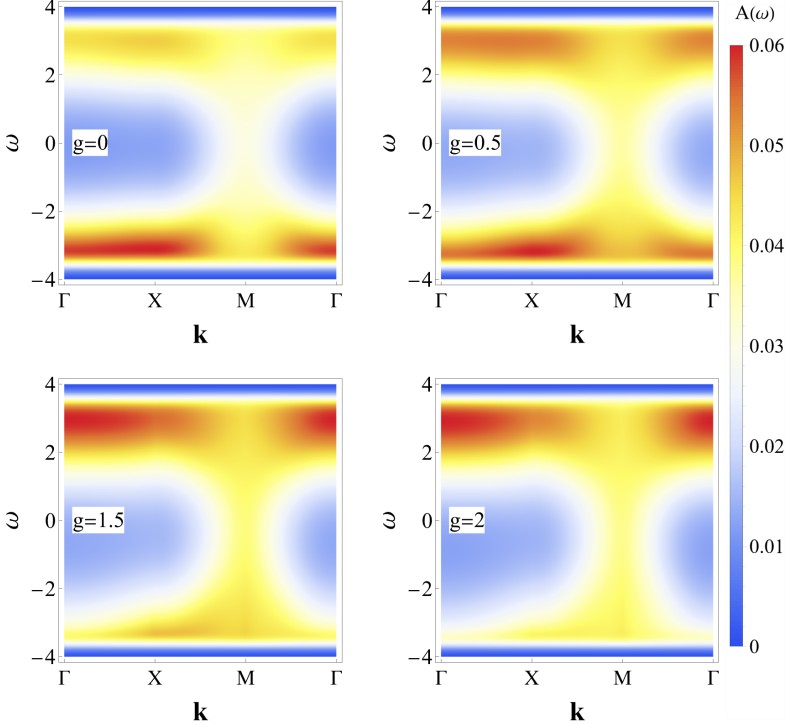

Figure 20: Semiholographic RPA Green function for four values of the hybridization $g$ and for holographic parameters $(\alpha, \Delta) = (-0.65, 1.3)$. The structure of the spectrum resembles a Mott insulator: no central peak but a (rather soft) gap, and Hubbard bands at finite $\omega$. This is a novel phenomenon in semiholography, however what happens is that the zero in $G_R^{-1}$ (coming from the pole in the holographic propagator $G_R$) cancels the zero in the free fermion propagator. This is not what happens in true Mott systems, where the self-energy develops a pole.

in the holographic propagator $G_R$ will approximately cancel each other in some interval of $g$. So what happens is that self-energy $G_R^{-1}$ develops a *zero* which cancels the zero in the kinetic term, rather than self-energy developing a pole, as it happens in true Mott insulators.

To conclude, the main message is that semiholography may serve as a good *fit* in $g$ to fully holographic Green functions, mimicking the true behavior quite well, however the physics behind it is different and cannot encapsulate true lattice effects.

Another approximation that we have used for a quick and dirty scan in the $(\alpha, \Delta)$ plane is to integrate the right-hand side and the boundary conditions of the EMD equations over a single unit cell $(x, y) \in (-1/(2Q), 1/(2Q))$. In this way we obtain a system of ordinary differential equations (because the derivatives over $x, y$ on the left-hand side also become trivial in this case) which we integrate using the same pseudospectral algorithm as for PDEs, but of course with just $\tilde{z}$ dependence the numerics is much easier. In the next step we write the stress-energy tensors and sources as the sum of the former (monopole or averaged) term and the quadrupole contribution (the dipole contribution is zero). This leads to the separation of variables and again results in a system of ordinary differential equations. Already the third (octupole) step results in a good approximation to the final solution. We have *never* used this method either for any of the production runs but we have used it to gain a quick glance over various corners of the parameter space. We will apply and describe in detail this scheme in a forthcoming work [76].

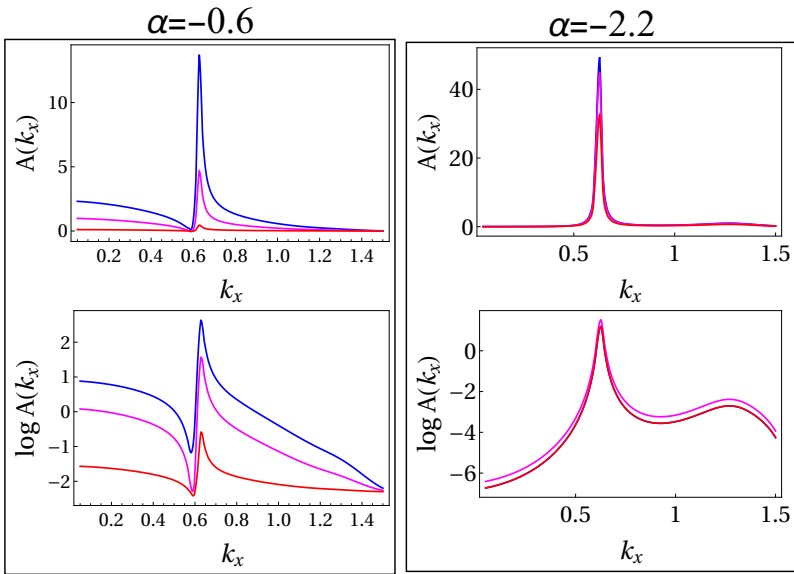

Figure 21: Three sections of momentum distribution curves (MDCs) $A(k_x, k_y)$ for $k_y = 0, 0.5, 1.0$ (blue, magenta, red), at $\omega = 10^{-3}$, for $\Delta = 1.4$ and $\alpha = -0.6$ (left) vs. $\alpha = -2.2$ (right). The curves capture a QP peak which exists for both cases; however the non-Fermi-liquid phase shows clear asymmetry of the peak, as opposed to the Fermi-liquid-like phase where the MDC peak is apparently Lorentzian.

## D  A quick look at momentum distribution curves

We give here a few remarks on the difficult subject of MDC structure. Because of the interplay of the lattice and the Fermi surface shape and the magnitudes of $\mathbf{k}_F$ and the lattice period, the phenomenology of MDCs shows vast variety. We thus limit ourselves to just pointing at what MDCs look like for Fermi-liquid-like regime as opposed to the non-Fermi-liquid regime. We work solely at $\kappa = 0$ now.

In Fig 21, we show the difference between a typical MDC in the non-Fermi-liquid phase (left) and in the Fermi-liquid-like phase (right). In both cases the system has a quasiparticle (as we can see also from the "phase diagram" in Fig. 4). However, the asymmetric structure of the QP peak, the telltale sign of non-Fermi-liquid physics, is clearly present in the left, whereas the right-hand plot has a symmetric peak. Notice that in this case the asymmetry of the left-side plots is seen also in EDCs (e.g. Fig. 5), which is expected for the holographic model of our type, since it is known that both in homogeneous space [27] and in the presence of weak lattice (Section 4.2) our choice of the EMD action implies that overdamped peaks are always asymmetric also in energy. But MDCs provide a direct and general piece of evidence of the non-Fermi liquid physics.[24]

As a taste of the work to be done, we present in Fig. 22 three two-dimensional MDC plots, again near zero energy, roughly corresponding to the three cases of EDCs in Fig. 5: a sharp but asymmetric quasiparticle, a broad continuum with no quasiparticles, and a more symmetric, Fermi-liquidish quasiparticle. Unusually, sharp EDC for $\alpha = -0.7$ becomes a broad MDC, whereas a complete lack of quasiparticle for $\alpha = -1.0$ translates to strongly localized (though of course finite in amplitude) MDC curve. The third case ($\alpha = -2.2$) is sharp both in momentum and energy. This potentially very intriguing phenomenon will be the subject of further work.

---

[24]We thank an anonymous referee for reminding us of this criterion.

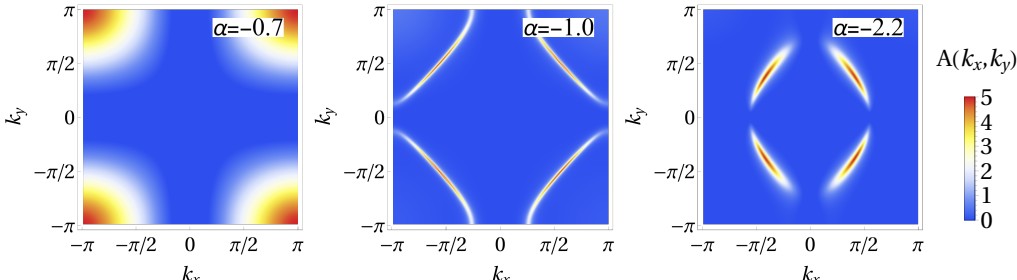

Figure 22: Two-dimensional plots of momentum distribution spectra $A(k_x, k_y)$ at $\omega = 10^{-3}$, for $\Delta = 1.3$ and $\alpha = -0.7$, $\alpha = -1.0$, $\alpha = -2.2$ (left to right). The first case has no sharp QP in momentum despite a sharp EDC. The second case is opposite to that: sharp momentum distribution even though we have shown that no sharp QP in energy exists. The final case looks close to a Fermi liquid.

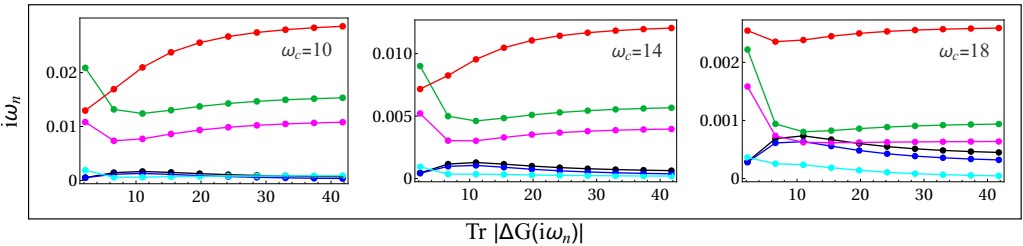

Figure 23: Absolute trace of the relative difference $\text{Tr}|\Delta G(i\omega_n)|$ for three cutoff values $\omega_c$, with respect to the largest cutoff we have tried $\omega_c = 22$. The spectra are computed for the six standard momentum values. Overall agreement is good, even for the largest cutoff employed.

# E  More on comparison to the Hubbard model

In this Appendix we discuss the practicalities of performing the Hilbert transform (57) numerically, we give some more data comparing the Green functions of the holographic lattice to those of the Hubbard model, and finally we show that the Hubbard-like bands are not present in absence of the lattice even with the dipole coupling $\kappa$ turned on. In other words, the band require both the dipole coupling and the lattice.

## E.1  Numerical implementation of the Hilbert transform

The numerical integration over frequencies in (57) has two nontrivial points: (1) the large-frequency limit $\omega \to \pm\infty$ (2) the vicinity of the QP peak. The first problem is resolved by imposing a large but finite cutoff so we really compute the integral $\int_{-\omega_c}^{\omega_c}$. The convergence is quite rapid already for $\omega_c \approx 10$ because of the boundary source $\mathcal{D}$, introduced in Eq. (16) and motivated in subsection 3.3, which acts as a UV regulator. One can estimate the UV convergence by comparing the results for different cutoff values. We have considered the cutoffs $\omega_c = 10, 14, 18, 22$. We have adopted the $\omega_c = 22$ results as reference values $G^{(0)}(i\omega_n)$ for the Matsubara propagator and calculated the relative difference $\Delta G(i\omega_n) \equiv |\left(G^{(i)}(i\omega_n) - G^{(0)}(i\omega_n)\right)/G^{(0)}(i\omega_n)|$, where the indices $i = 1,2,3$ correspond to $\omega_c = 18, 14, 10$. In Fig. 23 we show the values of the relative difference for the three cutoff values, for the standard set of momenta and for randomly chosen parameter values $\alpha = 1.1, \Delta = 1.3$. We see that already for the smallest cutoff the relative difference is about one percent, thus our results are reliable.

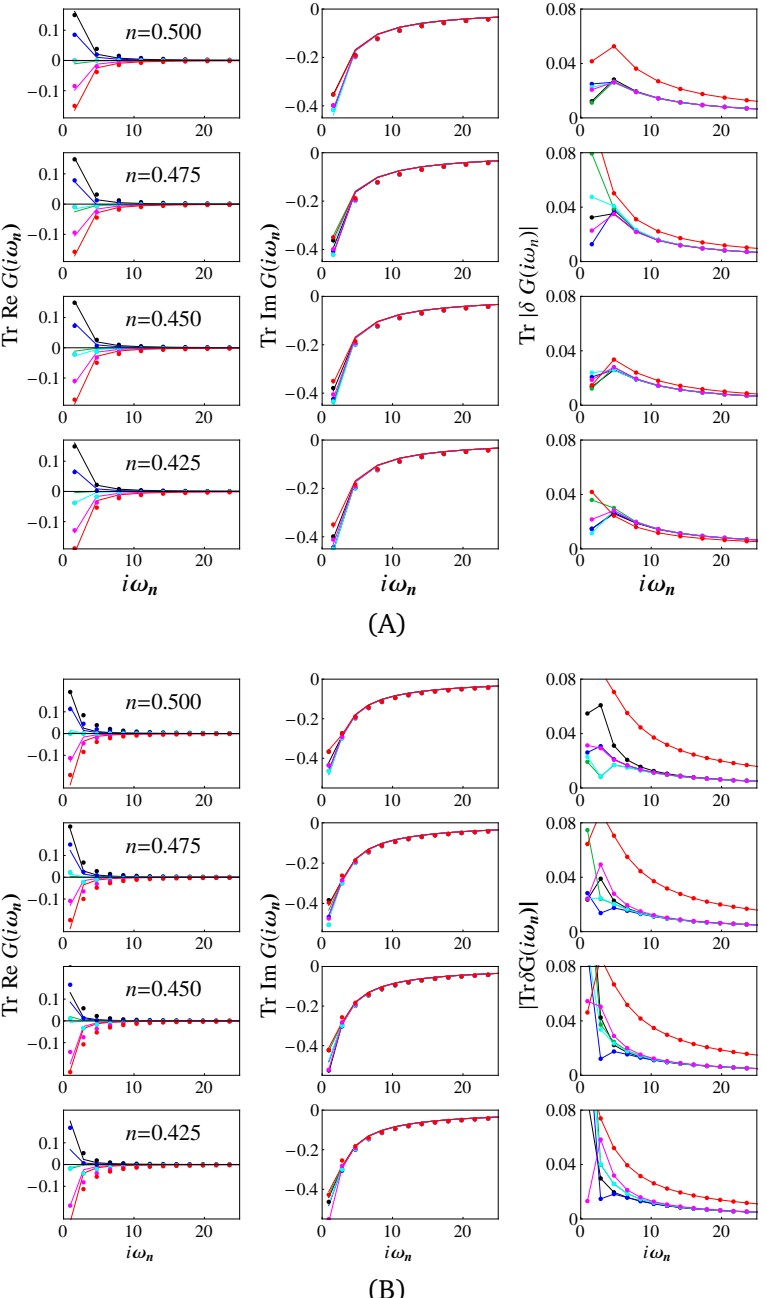

Figure 24: Same as Fig. 15 but for the temperature $T = 0.5$ (A) and $T = 0.3$ (B). At $T = 0.5$ the agreement is still quite good. For $T = 0.3$, the lowest temperature in the set of CTINT data that we have used, we see some systematic differences between the curves for small $i\omega_n$, which is absent on higher temperatures. This suggests differences in local features in real frequency, not really a surprise since the two models are microscopically distinct.

The second issue discussed above is essentially an IR problem: if the quasiparticle peak is very sharp, we need a much finer resolution when integrating near the peak; this is important as otherwise we do not get good large-$i\omega_n$ asymptotics. However, this is easily resolved by decreasing the integration step for by a factor $10 - 20$ near the peak maximum. Since this interval is very narrow (smaller than $\sim 0.01$), this presents no problems (the slowdown near the peak is significant but only constitutes a small part of the overall integration interval).

## E.2 Additional data on fitting the holographic spectra to the Hubbard model

Now we give the comparison of the Matsubara Green functions for the holographic and for the Hubbard model for the remaining two temperatures, $T = 0.5$ and $T = 0.3$ (Fig. 24). At $T = 0.5$ the agreement is still very good but at the lowest temperature, $T = 0.3$, there are larger systematic differences. The agreement is the worst at small Matsubara frequencies, which correspond to local features of real-frequency $G_R$ (large Matsubara frequencies correspond to global features, or properties of the integral $\int d\omega A(\omega, \mathbf{k})$). This simply means that detailed properties of the spectrum (which are best visible at low temperatures) differ between the AdS and Hubbard models. In the context of all our findings, this is expected. More interesting is the clear outlying status of the curve at $\mathbf{k} = (\pi, \pi)$, in the corner of the Brillouin zone. We have discussed in our WKB analysis why such points are expected to show diminished quasiparticle intensity. We thus understand the phenomenon in AdS but obviously it is something which does not happen in the Hubbard model.

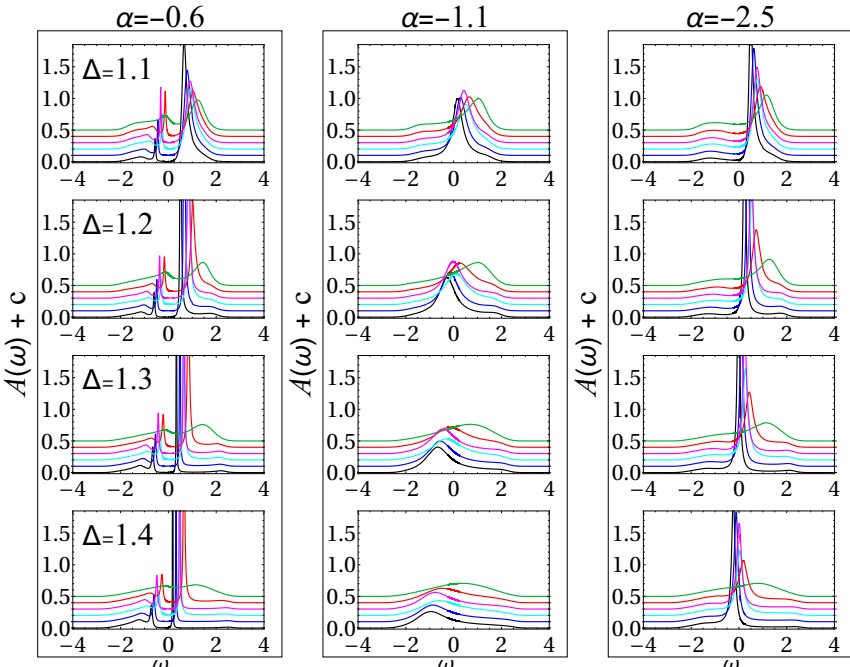

Figure 25: Energy spectra for the same values of $\alpha, \Delta, \kappa, \mu$ as Fig. 8 but in absence of lattice: $\delta\mu/\mu_0 = 0$. The gaps separating the left and right band from the central (quasiparticle) peak have almost vanished, in accordance with our expectation that the bands are well-separated thanks to the contributions of multiple Brillouin zones in the self-energy. The six momenta are chosen as $k/\mu_0 = 0, 0.5, 1.0, 1.5, 2.0, 2.5$.

## E.3 Absence of Hubbard bands in a homogeneous system

We will now present numerical evidence that in absence of lattice and the umklapp contributions to the self-energy the Hubbard bands are not present (or at least are much less prominent). We simply take the same values of other parameters but put $\delta\mu = 0$, i.e. we work in homogeneous space. As an example, consider Fig. 25, obtained with the same parameters as Fig. 8, including the dipole coupling $\kappa = 1.5$, except for the absence of lattice. In this case, the momenta can take any real values and the natural unit is the chemical potential $\mu_0$. While the overall structure of the spectrum does not differ too much from the weak lattice case for $\alpha = -1.1$ and $\alpha = -2.5$, there are no gaps and no band-like structures. For $\alpha = -0.6$, we

find the typical strong gap between two branches of the dispersion relation, both of them with sharp peaks; this is very different from the lattice result, and again has no broad bands (the only exception perhaps being the case $\alpha = -2.5, \Delta = 1.1$).

Note however that point 2 in subsection 4.2 provides an analytical argument for the same conclusion: that bands are a consequence of the umklapp terms in the Green function. Although that argument is clearly non-rigorous, it lends additional merit to the numerical reasoning in this Appendix.

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
