# Peer review of "Photoemission "experiments" on holographic lattices"

_SciPost Physics Core, doi:SciPost Phys. Core 6, 027 (2023)_

## Round 2 · Referee Report · Anonymous (Referee 1) · 2022-9-19

Report

The manuscript studies the single-electron spectral in a 2D holographic ionic lattice with hyperscaling-violating infrared geometry. As far as I know, this should be the first paper that computes the fermionic spectral function in a 2D holographic lattice. Some interesting results and robust features are observed. The structure of the spectral function is partially explained by a perturbative near-horizon analysis. Moreover, the authors compare the holographic results with the Hubbard model and find a good fit to the Hubbard Green's function, which brings AdS/CMT closer to the lab. The paper is clearly written and well-organized.

I have some questions and comments that are listed as follows.

  1. The authors imposed the Dirichlet boundary condition for the dilaton field $\Phi$ at the horizon. As mentioned in the last paragraph in subsection 2.1.1, they required $\Phi$ to drop to zero at the horizon $\tilde{z}=0$. I find no reason to set $\Phi(\tilde{z}=0)=0$. Instead, for the coordinate systems in Eq.(5), one can impose the Neumann boundary condition for $\Phi$, i.e. $\partial_{\tilde{z}}\Phi=0$ at the event horizon $\tilde{z}=0$.

  2. When introducing the spectral function $A(\omega, k)$, the authors disregarded all off-diagonal terms. But there are still an infinite number of diagonal terms. The authors should explain in detail how they define the spectral function in the holographic setup.

  3. In a density plot, a legend is necessary to illustrate the value of the physical quantity one considers. In particular, in Figure 7 and Figure 10.

  4. The authors claimed to set the source of the scalar operator to unity. More precisely, the UV expansion of $\Phi$ reads $\Phi(x,y,r)=r^{-\Delta_-}(\phi_s+\cdots)$ with $\phi_s$ the source of the dual scalar operator, and the authors wanted to set $\phi_s=1$. In terms of the new coordinate $\tilde{z}$ defined in Eq.(3), one has $\Phi(x,y,\tilde{z})=(1-\tilde{z}^2)^{\Delta_-}\left(\frac{\phi_s}{r_h^{\Delta_-}}+\cdots\right)$. Therefore, compared with Eq.(12), the authors actually fixed the scalar source $\phi_s= r_h^{\Delta_-}$ rather than $\phi_s=1$. Similarly, $\rho_(x,y)$ in the subleading term of $A_t$ in Eq.(12) should not be identified to be the charge density.

  5. I suggest the authors providing more details about their numerical methods.

1) How did they deal with the gauge fixing? DeTurck method or others? 2) Did they use double precision numbers or higher precisions? 3) Did the authors use High-Performance Cluster or desktop? 4) In Figure 18, the authors showed the running time $t$ in function of the size of lattice $N$. What’s the unit of $t$?

I recommend this manuscript for publication in Sci.Post, after the authors address the points raised here.

---

## Round 2 · Referee Report · Anonymous (Referee 2) · 2022-9-22

Report

The manuscript "Photoemission 'experiments' on hologrphic lattices" is devoted to the study of fermionic correlation functions in holographic models which incorporate the tunable sclaing behavior (via the running dilaton) and the square crystal lattice (via the modulation of the chemical potential). The obtained spectral functions are inspected and some of their features are noted. The analytic treatment of the ferionic self energy induced by the modulated horizon is provided. The comparison to the Green's functions calculated in Hubbard model is performed.

I see several notable original features of this work: 1) To my knowledge this is the first time when the calcualation of hologrpahic fermionic spectra is performed on top of the 2D lattice. This is a significant technical achievement, since it requires noumerical solution of 3D partial differential equations
2) This is the first attempt to perform a quantitative comparison of the holographic calculations with a more conventional Hubbard framework 3) The analytical perturbative treatment of self energy is an interesting result as well, although this can be seen as a relatively straightforward generalization of [4].

The amount of effort invested in this work is remarkable and it is of no doubt that the authors demonstrate the ability to handle quite complex technical methods in holographic lattices. However the analysis of the results of these calculations performed in the manuscript doesn't really lead to any significant physical conclucsions. The features noted in the spectral functions are quite generic, the comparison to Hubbard model is qualitative at best and the authors admit themselves that the merit function which they use for the fitting isn't really selective in the space of model parameters, so it dosn't tell much. There are many unsupported claims in the manuscript, which require a much more detailed analysis to be truly justified, as I list below. The validity of several arguments is questionable, as I also describe below. Because of this lack of clear and justified results, I can't recommend the current manuscript for publication in SciPost.

The unstructured list of major issues in the manuscript.

1) The authors don't specify the units in which they measure temperature, chemical potential, frequency etc in the hologrpahic model. This makes it impossible to judge the results and compare with the existing literature. The authors use the statement "low temperature" what is the meaning of it?

2) Despite the fact that the lattice is listed as the main feature of the current study, its effects are never really discussed in the manuscript. The spectral functions in the (kx,ky) plane are never shown and the geometry of the Fermi surface is not discussed. It is unclear what is the geometry of the energy bends and to what extend is it due to the umklapp effects and the hybridisation of the neighbouring Brilluin zones. The relation between the lattice constant and the size of the Fermi surface is unclear. I wonder, whether the "Hubbard bands" which are the main focus of the manuscript would not be observed already in the homogeneous model in absence of the lattice. We know that the fermionic spectral functions can be quite complicated in holography, featuring the nested Fermi surfaces the gaps, the smeared continuous density and vague bumps even in the hologeneous models. So in order to claim that "Hubbard bands" are the feature of the lattice, it should be demonstrated that they disappear in absence of the lattice.

3) The fermionic calucalations are quite nontrivial in presence of the lattice, as the authors discuss on p.12. (Btw, the value of Bloch momentum k either appears or disappears in the argumens in eqs. (18)-(21), looks like there is a confusion here). I think more details on the actual calculation procedure should be provided, in order to evaluate what exaclty is measured here. How the numerical boundary conditions are imposed? To which Bloch momenta belong the sources and the responces in the fermionic computation and how is it related to the Green's function element $G^{nn}$? How the calcualtion setup is organized for the cases of different scaling dimensions of the fermionic probe? These details are valuable and should be included in the Appendix. It is also very interesting how the authors obtain the momentum integrated spectral functions in Figure 11. Naively evaluating spectral function at al momenta would require enourmous computational effort.

4) When analysing the EDC spectra the authors use very vague definitions of the features they are talking about. It really doesn't go beyond visual inspection. How exactly do we distinguish between quasiparticle and non-quasiparticle? What do we mean by "exponentially suppressed by temperature" for temperatures of order 1. On Fig.6 (bottom right) I can discern three soft bumps, but the authors claim the structure is "unimodal", how is this defined? All this analysis doesn't seem robust.

5) In the paragraph below Fig.8 the authors use the terminology "underdoped", "optimally doped" etc. without any definition. By no means these term have a commonly accepted meaning in holographic models. Similarly, the definition of "Mottness" is never provided.

6) In the last paragraph before 2.1.2 it is said that dilaton is zero at the horizon. I think this is a misprint.

7) On the bottom of Fig.3 and the related analysis the values of the metric functions at $\tilde{z} = 1$ are shown. (Note the mistake in AxesLabel there). This is the asymptotic boundary of AdS, where the Dirichlet boundary condition is imposed. So it has no relevance to the discussion about lattice being absent in IR. Something is confused here.

8) Before Fig.5 there is the statement that the asymmetric peak in EDC signals the non-Fermi liquid behavior. I don't think this is correct. This statement should rather be aplied to MDC (momentum distribution curve) where Fermi liquid does indeed always give a Lorentzian.

9) In point 3 in the end of Sec. 4.2 there is a claim that the local maxima in the self energy $\Sigma$ explain the appearence of the bumps in the spectral function ("Hubbard bands"). I don't see how it can be true, since the location and the number of the peaks in the spectral function is controlled by the poles of the Green's function, whose position in turn depend on the bulk physics. Self energy, evaluated in Sec. 4.2 at the horizon only affects the width of these peaks.

10) In point 2 in the end of Sec. 4.2 there is a discussion that at the Brilluin zone boundary the numerator (btw it reads "numerators" in the text) in (53) goes to zero. I don't see how it works, since at the BZ boundary the Bloch momentum $k$ must be finite, of course. What it really meant here in this paragraph?

11) I wonder, how the integral in (55) is evaluated using the holographic results? Is there a cutoff imposed at some $\omega$? This calculation deserves being mentioned in the Appendix.

---

## Round 3 · Referee Report · Anonymous (Referee 2) · 2023-1-17

Report
In the revised manuscript the authors have taken into account my previous comments and improved the discussion in several aspects, which definitely clarifies the content of this work. The manuscript looks now like a solid and complete work. There are still a few points from my list, which have been overlooked (see the summary below), together with a few minor remarks, but they don't play a significant role.
The changes, while filling the holes in the presentation and argument, don't affect the overall physical output, which is (as I summarized in my previous report) to my mind still quite limited. So I think the manuscript in the current shape fully deserves a publication in SciPost Physics Core, but doesn't reach the impact of SciPost Physics.
Minor remarks/overlooked points: - Again, what are the units in the caption of Fig.1, which states Q=1/2? This contradicts the statement that all dimensionful quantities are measured in units of Q. - I don't really get the importance of the bottom row of Fig.3. As I understand, on the asymptotic boudary one sets Dirichlet boundary conditions for the metric fields. So what is plotted here is the value of the boundary conditions imposed and has no extra information. - As far as I understand the symmetric peak in the EDC means that the self-energy is linear in frequency at most. This is not true even for the plain Fermi liquid, where the self energy is frequency squared. - not addressed: How the authors obtain the momentum integrated spectral functions in Figure 11? Naively evaluating spectral function at al momenta would require enourmous computational effort. - In the analysis of the "bumps" in the spectral function in point 3 on p.28 I would expect the authors to analyse the expression for the imaginary part of the propagator (52), rather then the self energy itself. The connection is there, for sure, but it's still not clearly stated in the text.

---

## Round 3 · Author Response

Dear Editors, dear Referees,
We present now the updated version of the paper. We have done our best to address the questions in the reports. We have also included numerous minor clarifications and corrected a few typos. The overall methodology and message of the paper however stay pretty much the same as in the first version.
Response to Referee 1:
01) Indeed, the dilaton does not drop to zero at the thermal horizon, this is only true of $A_t$, as can be seen also from our Eq. (8) and the Appendix A. In the previous version we had a misprint attributing this behavior also to $\Phi$ (the actual IR boundary behavior is correct, both in Eq. (8) and further in the Appendix). In general, we obtain the boundary behavior of the dilaton from the IR expansion given in the main text and Appendix, and in the $\tilde{z}$ coordinate that indeed gives the Neumann condition, i.e. vanishing derivative. This is now corrected at the end of Section 2.1.2.
02) We have now expanded the Section 3.2 to include some more details of the Green function calculation (see also the point 03 in the response to the other referee).
03) We have now included a legend in all density plots, and in a few other (line) plots where we found it useful. We have not included a legend in the many EDC plots at six standard momenta because these momenta are introduced at the beginning and stay the same in all figures; we think including this one and the same legend every time would unnecessarily cram the plots. Also, we do not include the color legend in 3D plots in Fig. 1 and 3 as the colors are there just to guide the eye and the z-axis values can be directly read off from the frame ticks. We have mentioned this in the figure captions.
04) True, in the usual $r$ coordinate the source is rescaled by $r_h^{\Delta_-}$. The text is now more precise on this matter (it says we set to unity the leading term, which is proportional but not equal to source). This is not a problem however as the important point is to set a nonzero source so we do not form a condensate; this remains true no matter how large the source is.
It is also true that the subleading term of $A_t$ in Eq. (12) is not the charge density because the presence of the lattice makes the expression for charge density in terms of the derivatives of $A_t$ more complicated. We have thus renamed the coefficient to $a(x,y)$ and do not call it charge density anymore. Importantly, this change does not affect anything in the paper as we only tune the leading term ($\mu$) and do not work with the subleading term at all.
05) We have slightly expanded Appendix B so we discuss some more details of the numerics, in particular (1) we use the Einstein-DeTurck trick (2) we use double precision (3) for most runs we have used a cluster; for testing the code and for low-resolution spectra we have used a desktop computer (4) the running time of the test example is in seconds (now mentioned in the caption of Fig. 18).
In addition to above, we have also corrected a number of small inaccuracies and typos and clarified the text in a few places.
Response to Referee 2:
01) All dimensionful quantities are expressed in terms of the lattice wavevector $Q$. This is now stated at the very beginning of Section 4.
02) Concerning the Hubbard-like bands, we have now shown numerically (by computing the spectra for the same EMD model in homogeneous space) in Appendix E.3 that the Hubbard bands vanish in absence of lattice. This also follows from the analysis in point 3 of Section 4.2 (see also point 09 in this response).
Concerning MDCs, we agree that this is the more interesting part, however it requires substantial additional calculations and we believe it requires a separate work, in particular since the most interesting results seem to happen for strong lattices for which the numerics is very difficult (and goes beyond the scope of this paper where we work exclusively with weak lattices). We include a new Appendix (Appendix D) where we present a few examples of MDC curves, however no attempt is made for a systematic discussion.
03) Indeed the $\omega$- and $k$-dependence of the Bloch functions was not always spelled out explicitly in Eqs. (18-21), this is now corrected: we write the subscripts $\omega$ and $\mathbf{k}$ whenever the wavefunctions depend on them, whereas coordinate dependence is written as argument of a function (e.g. $\psi_{\omega\mathbf{k}}(x,y,z)$).
In Section 2.2 we have added a paragraph explaining in more detail the calculation of the Green function. In short, the source is a plane wave and from the response we read off the components from different zones so we can construct the diagonal component of the Green function (see also the point 02 in the response to the other referee).
04) Concerning the quasiparticle detection, the ultimate criterion (in absence of good analytic insight) is to compute the spectral function, i.e. the retarded propagator in the bottom half of the complex $\omega$ plane -- then one can directly see the pole and differentiate it from a branch cut. However, such a calculation was too demanding for our present work (it requires a large number of points and the convergence becomes progressively worse as we increase the magnitude of the imaginary part of $\omega$). Therefore, we have simply scanned the peak in real $\omega$ with increasing resolution, and decided we have the quasiparticle if the peak becomes ever sharper with resolution until its width drops below $O(T)$ at temperature $T$; if it is broader than the temperature times a factor of order unity then it is not a quasiparticle. This is not quite satisfying but is apparently the only choice in absence of analytic results. This is now explained in footnote 14.
Concerning the exponential suppression of the spectral weight inside the gap, more precise statement is that the gap is exponentially small compared to the ratio of the peak maximum to the typical "background" value of the spectral function, i.e. its value away from the peak (which in general grows with temperature). We have included this clarification in the text.
Concerning the "unimodality" of the high-temperature spectrum in Figs. 6 and 7, we agree it is not the right word. What we meant is really that the side bumps (bands) also melt away with $T$, not only the quasiparticle. This is now explicitly said in the text.
05) We agree that the "underdoped/overdoped" terminology is not uniquely defined and can be confusing. For that reason these terms are not used anymore. Instead, in relation to Fig. 11 (the only place where this terminology was used) we just directly describe the situation: the peak of the spectral weight shifts from $\omega<0$ (electrons) to $\omega\sim 0$ (excitations near the Fermi surface) to $\omega>0$ (holes). Mottness is defined as the shift of the spectral weight from high to low frequencies when the system is doped, without closing the gap, i.e. without entering the metalic phase.
06) Indeed, we had a typo, the dilaton does not drop to zero at the horizon. This is now corrected (see also the point 01 in the response to the first referee).
07) Indeed the axes label was wrong in the bottom panels of Fig. 3 (instead of x-z the axes are really x-y). Also, the text in the figure caption weas misleading, this is now corrected. The bottom figures are really just sanity checks that we get AdS asymptotics.
08) While in general it is indeed the asymmetric MDC which provides a definite proof of non-Fermi-liquid behavior, in EMD models of this type (with scaling solutions at zero temperature), it is known from the literature that non-Fermi-liquid phases always have asymmetric self-energies. This extends to our weak-binding analysis too. So while not true in general, it is true for this model. Also, in Appendix D we include a few MDCs which indeed show asymmetric structure. This is now explained (and the Appendix D referred to) in footnote 15.
09) It is true that the (imaginary part of) self-energy does not affect the existence nor the position of the poles (quasiparticle peaks), but the wide bumps/bands are not quasiparticle peaks and do not correspond to poles. Higher values (bands) and lower values (the minima between bands and the quasiarticle) come solely from the finite part of the propagator, i.e. from the self-energy. We briefly discuss this at the end of point 3 of Section 4.2.
10) Indeed this point was not clear in the original text. Actually it has nothing to do specifically with the edge of the BZ. The point is that the numerator of I (Eq. 53) can fall off faster than the denominator (if $\vert\mathbf{k}\vert<1$) so even for small $\omega$ the self-energy (proportional roughly to $\exp(-2I)$) can stay of order unity, which explains the weakening of the peak in some cases. The text of point 2 in Section 4.2 is now reformulated accordingly.
11) We now discuss in some more detail the integral (55), mainly in a newly written section in Appendix D. In short, a cutoff is indeed imposed at large $\omega$ since we cannot numerically integrate to infinity, however we show that the result is very stable and practically independent of the cutoff thanks to the fact that our spectral weight falls off exponentially at large $\omega$, in accordance with the dynamical boundary source motivated in Section 3.3.
In addition to above, we have also corrected a number of small inaccuracies and typos and clarified the text in a few places.
We thank both referees for their instructive and insightful remarks and comments.
The authors

---

## Round 3 · List of Changes

We have introduced numerous small changes, clarifications and corrections. We do not list here every single change, only more substantial ones.
01) Corrected typo in Eq. (11). 02) Improved discussion of UV boundary conditions in Section 2.1.2. 03) Corrected the axis labels and captions in Fig. 3. 04) Improved notation in Section 3.1, in particular Eqs. (18-22). 05) Expanded discussion of the boundary conditions for $G_R$ in Section 3.2.1. 06) Added a short discussion on detection of quasiparticles and gaps in Section 4.1 (p. 18). 07) Added a short discussion on symmetry of quasiparticle peaks in Section 4.1 (p. 19). 08) Clarified discussion of Fig. 11 in its caption and in the text of Section 4.2.3 (p. 23). 09) Improved interpretation of results in Section 4.2 (p. 28-29). 10) Newly written Appendix D and Figs. 21-22 with some very basic remarks on MDCs. 11) Added Section E.1 and Fig. 23 on the numerical implementation of the Hilbert transform. 12) Added Section E.3 and Fig. 25 on absence of Hubbard-like bands in absence of lattice.
xx) Added a legend in Figs. 7, 10, 17.

---

## Round 4 · Author Response

Dear Editors, dear Referees,

We have made some minor revisions to the paper in accordance with the suggestions of the referees. A point-by-point response follows.

Referee 1:

Indeed, the index on the right-hand side of Eq. (72) was offset by 1, now it is corrected.

Referee 2:

1) In Fig. 1 we indeed have Q=1/2, i.e. all quantities are given in terms of the computational unit 2Q. At the beginning of Section 4 (i.e. after Figs. 1-3) we state that from now on, i.e. from page 16 onward, we always put Q=1. Hence there is no contradiction. 2) The bottom row in Fig. 3 is indeed just the sanity check for the numerics and the implementation of the boundary conditions. It has no physical significance except to remind the reader that the lattice is only sourced by the gauge field and not directly imposed in the metric itself. 3) In the vicinity of the Fermi surface, quadratic imaginary self-energy (and indeed any function which is analytic and even in ω) will certainly give a symmetric peak. Maybe the confusion stems from the fact that, further away from the Fermi surface, there is of course no reason that the peak be symmetric. We have now emphasized that this criterion holds only for reasonably sharp peaks, not far away from the Fermi surface. 4) Indeed we have missed this point in the previous revision. True, computing the spectral function on a very dense grid of momenta would be prohibitively expensive computationally. However, our resolution is not very high: we use a 4x4 grid of momenta (as is often done also in Quantum Monte Carlo calculations of lattice models), hence the curves in Fig. 11 are visibly not completely smooth. While adding more points would of course be advisable, the present result is good enough to show our qualitative conclusions. We have now added this information to the caption of Fig. 11. 5) The relation between the behavior of the self-energy and the spectral function itself is simple in this case (when we are not right at the pole): the larger the imaginary self-energy the smaller the spectral weight. We have mentioned this in the text now.

We hope that the revised manuscript is now acceptable for publication in SciPost Physics Core.

Kind regards,

the authors

---

## Round 4 · List of Changes

1) Corrected typo in Eq. (72).
2) Added emphasis that the symmetry of the peaks is only a good indicator of the character of the quasiparticle sufficiently close to the Fermi surface (footnote 16, page 19).
3) Expanded caption of Fig. 11 now states the spatial resolution in computing the local spectral functions.
4) Added a remark in point 3 at page 27 on the relation between the self-energy and the spectral weight.

---

## Editorial Decision

published